# sRNA-mediated activation of gene expression by inhibition of 5'-3' exonucleolytic mRNA degradation

Sylvain Durand[1]*, Frédérique Braun[1], Anne-Catherine Helfer[2], Pascale Romby[1,2], Ciarán Condon[1]*

[1]UMR8261 CNRS, Université Paris Diderot (Sorbonne Paris Cité), Institut de Biologie Physico-Chimique, Paris, France; [2]Université de Strasbourg, CNRS, Architecture et Réactivité de l'ARN, Strasbourg, France

**Abstract** Post-transcriptional control by small regulatory RNA (sRNA) is critical for rapid adaptive processes. sRNAs can directly modulate mRNA degradation in Proteobacteria without interfering with translation. However, Firmicutes have a fundamentally different set of ribonucleases for mRNA degradation and whether sRNAs can regulate the activity of these enzymes is an open question. We show that *Bacillus subtilis* RoxS, a major *trans*-acting sRNA shared with *Staphylococus aureus*, prevents degradation of the *yflS* mRNA, encoding a malate transporter. In the presence of malate, RoxS transiently escapes from repression by the NADH-sensitive transcription factor Rex and binds to the extreme 5'-end of *yflS* mRNA. This impairs the 5'-3' exoribonuclease activity of RNase J1, increasing the half-life of the primary transcript and concomitantly enhancing ribosome binding to increase expression of the transporter. Globally, the different targets regulated by RoxS suggest that it helps readjust the cellular $NAD^+$/NADH balance when perturbed by different stimuli.

*For correspondence: durand@ibpc.fr (SD); condon@ibpc.fr (CC)

**Competing interests:** The authors declare that no competing interests exist.

## Introduction

Post-transcriptional regulation is a rapid and efficient way to modulate gene expression used by all organisms. In bacteria, this regulation can involve *cis*-acting regulatory elements such as riboswitches, which bind small molecules and modify transcription elongation, translation or mRNA stability. It can equally involve *trans*-acting factors such as proteins (RNases or RNA binding proteins) and small regulatory RNAs (sRNAs). sRNAs can be encoded in cis (antisense RNA) or in trans relative to their target mRNAs and these regulatory mechanisms can be superimposed on each other to create complex control networks.

The two best-studied model bacteria *Escherichia coli* and *Bacillus subtilis* have evolved with different arsenals of ribonucleases and alternative approaches to the fundamental cellular processes of RNA maturation and degradation (*Bechhofer, 2009*; *Durand et al., 2015b*; *Condon and Putzer, 2002*). Only eight of the >30 ribonucleases currently identified in these two bacteria are present in both organisms. *E. coli* depends primarily on endonucleolytic cleavage by RNase E to initiate RNA turnover, followed by degradation of the resulting fragments in a 3'-to-5' orientation by exoribonucleases. *B. subtilis,* on the other hand, has replaced RNase E by an enzyme with similar properties called RNase Y and has the additional option of exonucleolytically degrading RNAs from the 5' end using an enzyme called RNase J1 (*Mathy et al., 2007*). In this regard, *B. subtilis* shares features with the eukaryotic model of RNA decay and this observation has obscured some of the traditional divisions between « prokaryotic » and « eukaryotic » pathways of RNA degradation.

Small regulatory RNAs have been mostly studied in Gram-negative bacteria, especially in *E. coli* and its pathogenic relatives. In these organisms, sRNAs generally act by an imperfect base pairing with their target mRNAs. The interaction between sRNA and mRNA often requires the chaperone Hfq, an Sm-like protein that both stabilizes the sRNA and facilitates the pairing with mRNA targets (for review, see *Updegrove et al., 2016*). The sRNA-mRNA interaction can lead to a positive or a negative effect on gene expression. In bacteria, most sRNAs examined to date directly affect mRNA translation with an indirect effect on mRNA half-life (reviewed in *Wagner and Romby, 2015*). However, some more recent studies have shown that sRNAs are also able to directly affect mRNA stability without affecting translation, by recruiting the degradation machinery or interfering with its action (reviewed in *Lalaouna et al., 2013*). In one particular example, the ternary complex formed between the sRNA (MicC), the mRNA (*ompD*) and Hfq can recruit RNase E to a site within the open reading frame and stimulate its cleavage by presenting the 5' monophosphate of the sRNA to RNase E (*Bandyra et al., 2012*). In another, the sRNA RydC increases the stability of the *cfa* mRNA by pairing to the 5'-UTR of this mRNA, far upstream of the ribosome binding site, and hiding a cleavage site from RNase E (*Fröhlich et al., 2013*).

Small RNAs have also been studied in some Gram-positive pathogenic bacteria such as *Staphylococcus aureus* and *Streptococcus pyogenes,* but in *B. subtilis*, the paradigm of Gram-positive bacteria, only a handful of sRNAs have been characterized to date and the global mechanisms of sRNA regulation have not been fully resolved (*Durand et al., 2015b*). In these bacteria, Hfq is generally not required for sRNA-mRNA pairing (*Hämmerle et al., 2014*; *Rochat et al., 2015*) although there are some exceptions in specific organisms (*Nielsen et al., 2010*; *Boudry et al., 2014*). One key question is to what extent RNases J1 and Y are influenced by sRNA binding to mRNA targets in *B. subtilis*, i.e. can *E. coli* models involving RNase E be transposed to these RNases in its Gram-positive counterpart?

RoxS (related to oxidative stress) is the unique *trans*-acting sRNA, apart from the ubiquitous 6S RNA, that is conserved between *B. subtilis* and the Gram-positive pathogen *S. aureus*. First identified in *S. aureus* where it is named RsaE, it was shown to regulate genes involved in amino acid and peptide transport, cofactor synthesis, lipid metabolism, carbohydrate metabolism and the TCA cycle (*Geissmann et al., 2009*). In a previous study, we showed that RoxS transcription is induced during nitric oxide (NO) stress by the two-component system ResDE and regulates the expression of numerous genes linked to oxido-reduction processes in *B. subtilis* (*Durand et al., 2015a*). In searching for additional direct targets of RoxS that may not have been expressed under the experimental conditions of the first study, we found that RoxS potentially interacts with the extreme 5'-end of the *yflS* mRNA, encoding a malate transporter in *B. subtilis*. We show here that RoxS expression is greatly increased in the presence of malate in the medium and acts as a positive regulator of transporter expression by directly blocking *yflS* mRNA degradation by the 5'−3' exoribonuclease J1. This is the first known case of an sRNA controlling 5'-exoribonuclease activity in bacteria. RoxS can further stimulate translation of the stabilized *yflS* mRNA, possibly by opening a specific fold of the mRNA that partially occludes the ribosome binding site, providing an additional layer of complexity to the control mechanism. Lastly, we show that RoxS expression is controlled by Rex, a transcriptional repressor of genes linked to fermentation processes that senses the $NAD^+$/NADH balance in the cell. The increase in RoxS expression in the presence of malate is explained by a release from Rex-mediated repression, allowing a far more efficient regulation of RoxS targets than seen previously.

## Results

In a previous study we showed that RoxS, either directly or indirectly, negatively regulates the expression of up to a hundred genes, including two mRNAs encoding proteins involved in central carbon metabolism: *ppnkB* (encoding an inorganic polyphosphate/ATP-NAD kinase) and *sucCD* (encoding succinate dehydrogenase) (*Durand et al., 2015a*). In both cases, RoxS binds to the Shine-Dalgarno (SD) sequences and inhibits translation initiation. To expand the number of direct RoxS targets, we searched for new potential binding sites for this sRNA (TargetRNA2; *Kery et al., 2014*) that we might have missed in the first study because of the specific growth conditions used (rich medium at 37°C). One such predicted target, the *yflS* mRNA (encoding one of at least four malate transporters in *B. subtilis*) drew our attention because RoxS was predicted to bind to the extreme 5'-end of the mRNA well upstream of the ribosome binding site (*Figure 1*).

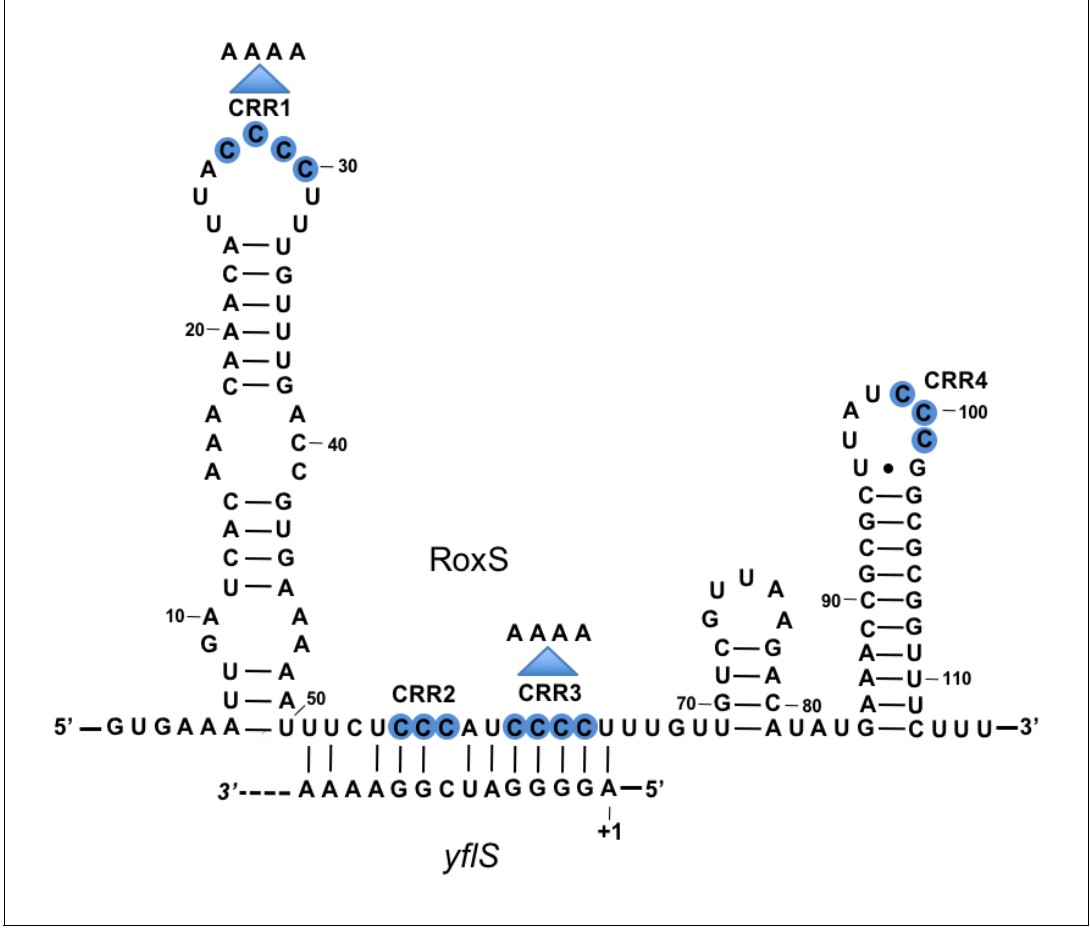

**Figure 1.** Base-pairing between RoxS and *yflS* mRNA predicted by the TargetRNA2 program (http://cs.wellesley.edu/~btjaden/TargetRNA2/). The transcriptional start (+1) of the *yflS* mRNA is indicated. This was mapped in *Figure 1—figure supplement 1* and its conservation is shown in *Figure 1—figure supplement 2*. The C-rich regions (CRR) of RoxS are in blue. Mutations CRR1 and CRR3 are indicated by arrows.

The following figure supplements are available for figure 1:

**Figure supplement 1.** Mapping of the *yflS* 5'-end by primer extension.

**Figure supplement 2.** Conservation of the *yflS* promoter and 5'-UTR in various *Bacilli*.

## Mapping of the 5' extremity of the *yflS* mRNA

The RoxS sRNA is predicted to base-pair to the extreme 5' end of the *yflS* mRNA transcribed from a putative −10 promoter element located 44 nucleotides (nts) upstream of the translational start codon. To confirm the existence of a 5' end corresponding to this promoter prediction and confirm it was a primary rather than processed transcript, we performed primer extension assays on total RNA with or without pretreatment with the 5'−3' exoribonuclease Xrn1. This ribonuclease only degrades RNAs with a 5'-monophosphate and leaves 5'-triphosphorylated primary transcripts intact. As a control, we used the *hbs* mRNA, for which two primary transcripts (P3 and P1) and a ribosome-protected maturation product (M) can be detected in vivo (*Daou-Chabo et al., 2009*). Treatment of total RNA by Xrn1 caused the 5'-monophosphorylated end corresponding to the *hbs* maturation product to disappear as expected. In contrast, the 5' ends corresponding to the P3/P1 primary transcripts of *hbs* and the predicted primary transcript of *yflS* were resistant to Xrn1 treatment (*Figure 1—figure supplement 1*). The primer extension product obtained with the *yflS*-specific oligonucleotide mapped to two consecutive residues (*Figure 1—figure supplement 1*) suggesting that the *yflS* mRNA can start at an A or G residue, 5 or 6 nts downstream of the predicted −10

region, and thus has a 38/39 nt 5'-UTR. This promoter and 5'-UTR is highly conserved among *Bacilli* with a clearly identifiable *yflS* gene (*Figure 1—figure supplement 2*). Since the RoxS sRNA sequence is identical to that of *B. subtilis* in these organisms, the predicted base-pairing with the 5' end of the *yflS* mRNA is likely to be conserved.

## RoxS regulates *yflS* mRNA stability

The *yflS* gene encodes a malate transporter whose expression is induced by the two-component system (MalKR), which senses the presence of malate in the medium (*Tanaka et al., 2003*). To determine whether RoxS had an effect on *yflS* expression, we compared the half-life of this mRNA in WT and Δ*roxS* strains grown in 2xTY containing 0.5% malate (*Figure 2A*, *Figure 2—figure supplement 1*). We detected a single band of approximately 1.5 kb corresponding to the predicted size of the full-length *yflS* mRNA on Northern blots of total RNA isolated at different times after addition of rifampicin to stop new transcription. The half-life of the *yflS* mRNA was 2.5 min in a WT strain but decreased to 0.7 min in the absence of RoxS. Thus, RoxS stabilizes the *yflS* mRNA about 3.6-fold, presumably by protecting it from degradation by cellular RNases.

## RoxS blocks degradation of the *yflS* mRNA by RNase J1

To identify which RNase(s) attack the *yflS* mRNA in the absence of RoxS, we measured the half-life of the *yflS* mRNA in strains lacking the three main RNases involved in *B. subtilis* mRNA degradation (RNase III, RNase Y and RNase J1). These experiments were done both in the presence and absence of RoxS (*Figure 2B–D*, *Figure 2—figure supplement 1*). Deletion of the gene encoding RNase III (Δ*rnc*) had no major impact on *yflS* mRNA half-life in either the presence (3.1 min) or absence (0.7 min) of RoxS (*Figure 2B* compared to *Figure 2A*). In the RNase Y (Δ*rny*) mutant strain, the full-length *yflS* mRNA was stabilized about 10-fold compared to WT (25.9 min *vs* 2.5 min half-life) and in the Δ*roxS* Δ*rny* double mutant compared to the Δ*roxS* single mutant (9.5 min *vs* 0.7 min). This result shows that RNase Y plays a major role in *yflS* mRNA turnover. However, the *yflS* mRNA is still destabilized about 2.7-fold in the absence of RoxS in the Δ*rny* mutant background (25.9 min *vs* 9.5 min half-lives, respectively, in the Δ*rny* and Δ*roxS* Δ*rny* strains). This suggests that RoxS does not protect the full-length *yflS* transcript from attack by RNase Y and that the effect of RNase Y on *yflS* turnover is RoxS-independent.

The deletion of *rnjA* gene, encoding the 5'−3' exoribonuclease RNase J1, led to a massive accumulation of stable degradation intermediates, presumably the products of endonucleoytic cleavage by RNase Y. The accumulation of these 3' fragments was similar in the presence or absence of RoxS, consistent with the observation (above) that RNase Y cleavages were RoxS independent. In the presence of RoxS, the stability of the full-length *yflS* mRNA was slightly higher in the Δ*rnjA* strain compared to wild-type strains (4 *vs* 2.5 min half-life, respectively). Remarkably, however, deletion of RoxS no longer destabilized the full-length *yflS* mRNA in the Δ*rnjA* background (4 *vs* 3.8 min half-lives in Δ*rnjA* and Δ*roxS* Δ*rnjA* strains, respectively). Thus, RoxS protects the full-length *yflS* mRNA from degradation by RNase J1 in vivo.

RNase J1 is known to be inhibited by the tri-phosphate group present on the 5' end of primary transcripts (*Mathy et al., 2007*). Direct attack of the 5' end of the primary *yflS* transcript by RNase J1 in 5' exonucleolytic mode would thus require conversion of the 5' tri-phosphate to mono-phosphate. To date, the only known enzyme to catalyze this reaction in *B. subtilis* is the RNA pyrophosphohydrolase RppH (*Richards et al., 2011*), although other as yet unidentified enzymes are thought to have similar activity. BsRppH prefers a guanosine residue in the second position of its RNA substrates (*Piton et al., 2013*). As the *yflS* mRNA fulfills this requirement (i.e. it starts with AG or GG), we measured the half-life of the *yflS* mRNA in both Δ*rppH* and Δ*roxS* Δ*rppH* strains (*Figure 2E*, *Figure 2—figure supplement 1*). In the Δ*rppH* strain, the half-life was similar to that measured in the wild-type (2.2 *vs* 2.5 min). The *yflS* half-life was also comparable between the Δ*rppH* Δ*roxS* and the Δ*roxS* strains (both 0.7 min). Thus, RppH is not the enzyme responsible for the removal of the 5' tri-phosphate of *yflS* mRNA to give access to RNase J1. This result is consistent with the idea that other RNA pyrophosphohydrolase activities are waiting to be identified in *B. subtilis.*

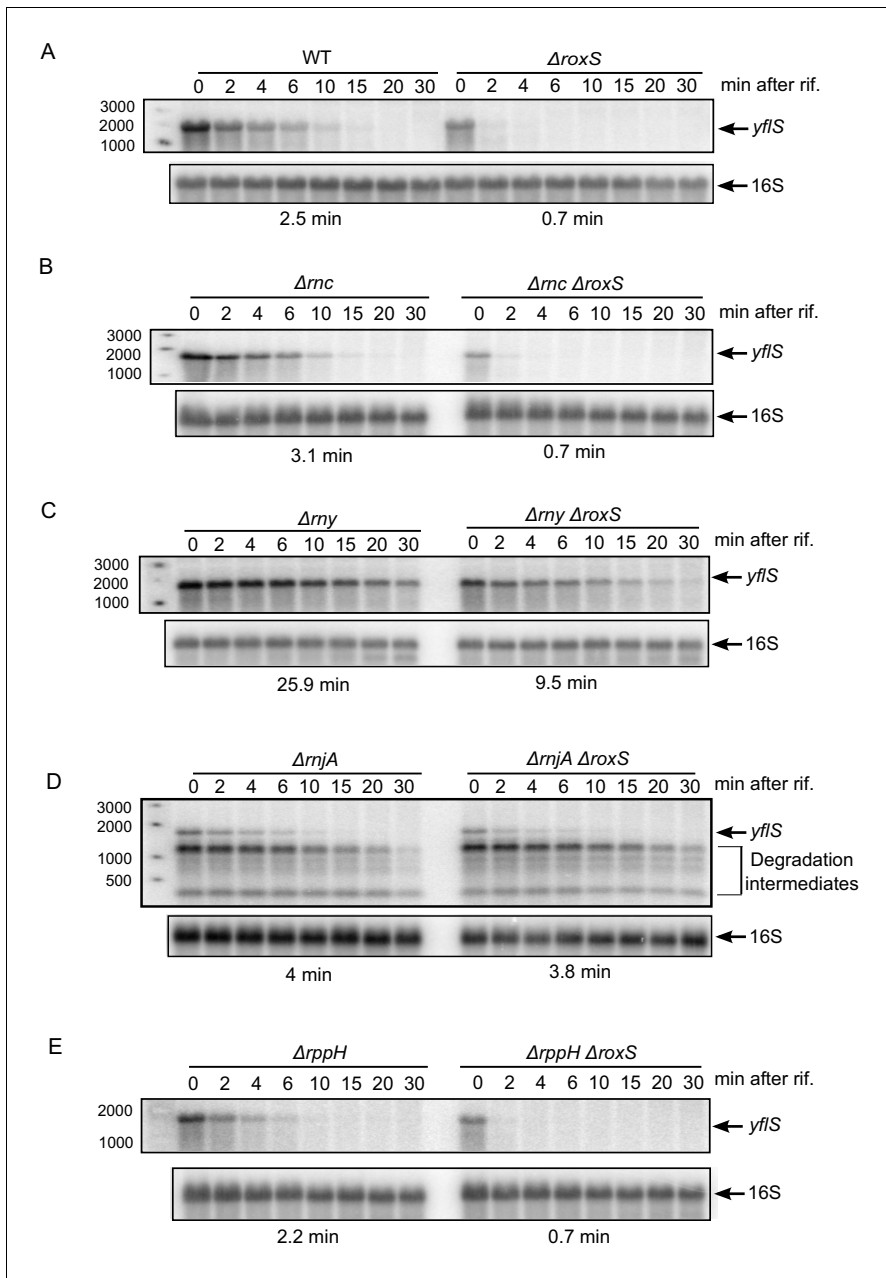

**Figure 2.** RoxS protects the full-length *yflS* mRNA from RNase J1 degradation in vivo. A representative Northern blot showing total RNA isolated at times after addition of rifampicin (Rif) to (A) WT and Δ*roxS* strains, (B) Δ*rnc* and Δ*rnc* Δ*roxS* strains, (C) Δ*rny* and Δ*rny* Δ*roxS* strains, (D) Δ*rnjA* and Δ*rnjA* Δ*roxS* strains and (E) Δ*rppH* and Δ*rppH* Δ*roxS* strains grown in the presence of 0.5% malate. Rifampicin was added at mid-exponential phase (OD$_{600}$ = 0.6). Half-lives calculated from two independent experiments (biological replicates) are given under each autoradiogram. The quantification of the Northern blots is presented in *Figure 2—figure supplement 1*.

The following source data and figure supplements are available for figure 2:

**Figure supplement 1.** RoxS protects the full-length *yflS* mRNA from RNase J1 degradation in vivo.

**Figure supplement 1—source data 1.** Source data for the quantification of *yflS* mRNA half-life in WT and *Bacillus subtilis* RNase mutant strains (Δ*rnc*, Δ*rnc* Δ*roxS*, Δ*rny*, Δ*rny* Δ*roxS*, Δ*rnjA*, Δ*rnjA* Δ*roxS*, Δ*rppH*, Δ*rppH* Δ*roxS*).

## RoxS protects *yflS* mRNA from in vitro degradation by RNase J1

The results described above suggested that RoxS protects the *yflS* full-length mRNA from attack by RNase J1 in vivo. To confirm this observation in vitro, we transcribed a fragment corresponding to the first 94 nucleotides of the *yflS* mRNA using T7 RNA polymerase and submitted this transcript to degradation by RNase J1 in the presence or absence of the RoxS sRNA. The RNA contained a $^{32}$P-labelled 5'-monophosphate group to mimic the in vivo situation following RNA pyrophosphohydro-lase activity. The *yflS* RNA fragment was attacked by RNase J1 in 5'-exoribonuclease mode as indicated by the accumulation of free radio-labelled AMP at the bottom of the gel, beginning after only 2 min of incubation with the enzyme (*Figure 3*, lanes 2–5). In contrast, almost no AMP was visible when the reaction was performed in the presence of RoxS, showing clearly that the 5' end of *yflS* is protected from RNase J1 by the sRNA (*Figure 3*, lanes 8–11).

## RoxS uses C-rich region 3 (CRR3) to interact with the extreme 5'-end of *yflS* mRNA

RoxS has four C-rich regions (CRRs) that can potentially base-pair with G-rich complementary sequences in its targets (*Figure 1*). In our previous work, we showed that RoxS mainly uses CRR3 to regulate the expression of the *ppnkB* mRNA (*Durand et al., 2015a*). To determine which CRR was most important for the interaction with *yflS*, we repeated the RNase J1 protection experiment using three different mutants of RoxS: one in CRR1, one in CRR3 and one combining both mutations (CRR1+3) (*Figure 1*). The RoxS CRR1 mutant still protected *yflS* against RNase J1 (*Figure 3*, lanes 14–17), suggesting that this region of RoxS is not important for the interaction with *yflS*. In contrast, the *yflS* RNA was no longer protected by the CRR3 RoxS mutant (*Figure 3*, lanes 20–23) or by the double CRR1+3 mutant (*Figure 3*, lanes 26–29). These results suggest that the CRR3 region of RoxS is the key C-rich motif for the initiation of base-pairing and protection of the 5' end of the *yflS* mRNA.

To confirm this result, we made compensatory mutations in the CRR3 of RoxS (CRR3-2) and at the 5' end of *yflS* (*yflS*$^{mut5'}$) (*Figure 4A*) and assayed the capability of these variants to protect *yflS* from RNase J1 degradation in vitro. The RoxS CRR3-2 mutant showed significantly weaker protection of the *yflS*$^{WT}$ transcript than wild-type RoxS (*Figure 4B,C*). Even though the *yflS*$^{mut5'}$ RNA was intrinsically a better substrate for RNase J1 than *yflS*$^{WT}$, wild-type RoxS similarly showed weaker protection of the *yflS*$^{mut5'}$ RNA than the CRR3-2 compensatory mutant (*Figure 4B,C*). These experiments strongly support the predicted base-pairing between RoxS CRR3 and the 5' end of the *yflS* mRNA.

Lastly, we performed structure probing of RoxS in the presence and absence of *yflS* using the RNase V1 (specific for double stranded RNA) and RNase T2 (preference for unpaired adenines). The cleavage sites were in good agreement with the secondary structure of RoxS, as most of the strong RNase T2 cuts were located in the apical loops, while the RNase V1 cuts were primarily located in the helices (*Figure 4—figure supplement 1*). The addition of *yflS* RNA caused limited changes in reactivity restricted to the region encompassing CRR2 and CRR3. Several RNase V1 cleavages were induced close to CRR3 (at nts 63–65). Concomitantly, several RNase T2 cleavages located between CRR2 and CRR3 decreased under the same conditions. Together, these observations show convincingly that, as for the *ppnKB* mRNA, RoxS primarily uses the CRR3 region to hybridize to *yflS* mRNA.

## RoxS also activates translation of the *yflS* mRNA

The data presented thus far suggests that RoxS binds to the 5' end of the *yflS* mRNA principally via the CRR3 motif to protect it from degradation by RNase J1. To determine the impact of RoxS binding on the conformation of the *yflS* mRNA and potentially on translation, we also probed the secondary structure of the leader portion of *yflS* alone or bound to RoxS. The probes used in this case were RNase V1, RNase T1 (specific for unpaired guanines) and RNase T2. A summary of the results is presented in *Figure 5*. In the absence of RoxS, two equilibrium secondary structure models best explain the enzymatic cleavage patterns detected. In the two proposed models, the SD and the AUG start codon are engaged in weak basepairing interactions, while the four guanines at the extreme 5' end of the *yflS* mRNA are accessible to RNase T1 and are therefore in single-stranded conformation. However, we noticed that adenines 11 to 14 were cleaved poorly by RNase T2 compared to other A-stretches (A50-54, A63-66, *Figure 5—figure supplement 1*). Their three-

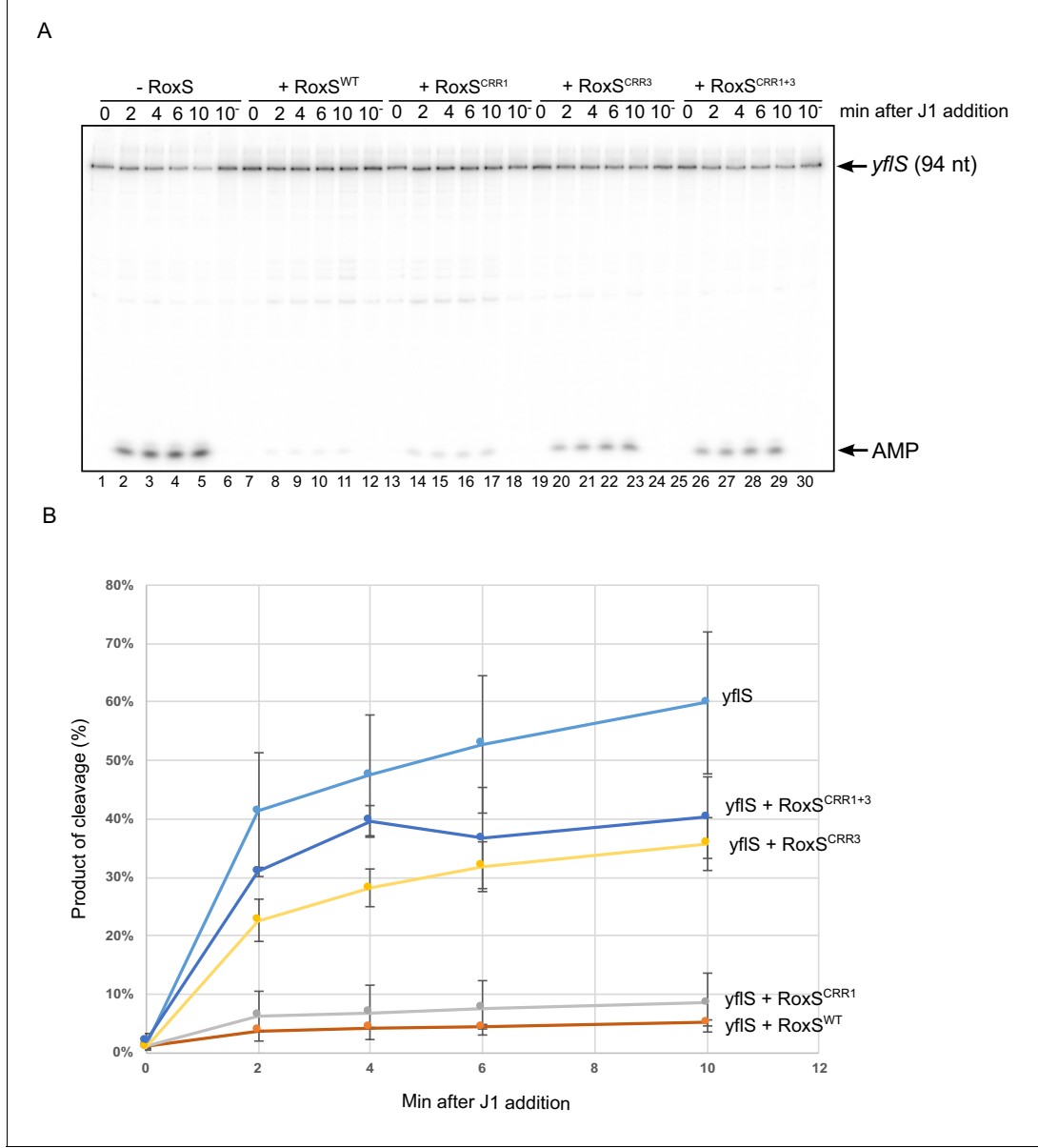

**Figure 3.** RoxS protects the *yflS* mRNA from RNase J1 degradation in vitro. (**A**) In vitro degradation of the 5' end of *yflS* RNA by RNase J1 ± RoxS sRNA or RoxS CRR mutants (see *Figure 1*). The *yflS* mRNA was in vitro transcribed, dephosphorylated with CIP and radiolabeled at its 5'-end to obtain a 5' monophosphate extremity. Times after RNase J1 addition are indicated at the top of the gel. Reactions were done at 37°C. The migration positions of the *yflS in vitro* transcript (94 nt) and AMP are indicated. (**B**) Percent release of AMP from *yflS* substrate RNA by RNase J1 ± RoxS sRNA or RoxS CRR mutants as a function of time (minutes). The data shown correspond to the quantification of two independent experiments (technical replicates). See also *Figure 3—source data 1*.

The following source data is available for figure 3:

**Source data 1.** Source data for the quantification of *yflS* degradation by RNase J1 in vitro.

dimensional conformation may therefore be more complex than shown here. Upon addition of RoxS, we observed a significant decrease in the RNase T1 cleavages in the first 10 nucleotides of the *yflS* leader, while adenines 11 to 14 become more accessible to RNase T2. A similar effect was seen with the RoxS CRR1 mutant. However, no major effect was observed with the CRR3 variants, consistent with the idea that CRR3 is the key mediator of the interaction with the 5' end of the *yflS* mRNA. In

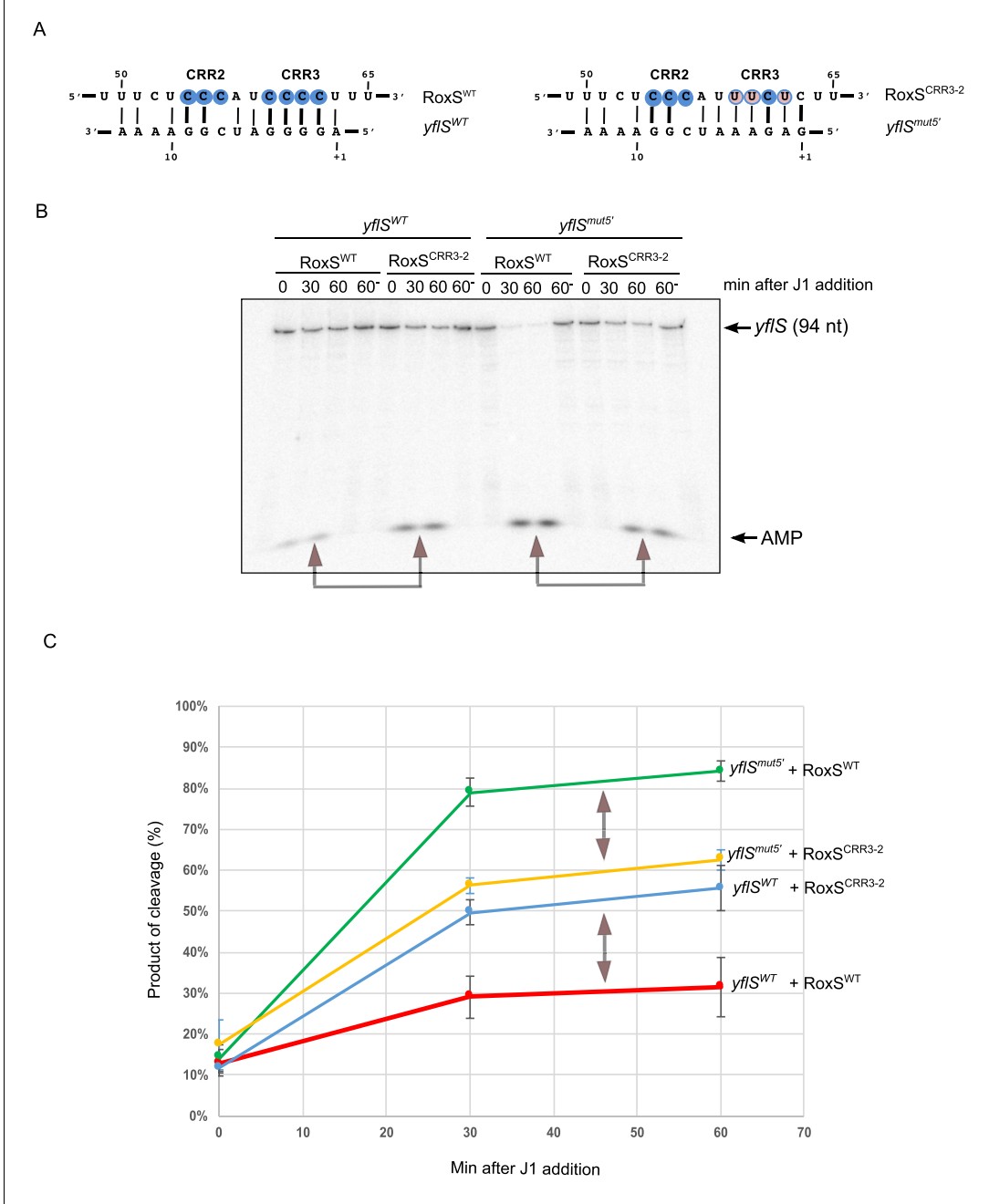

**Figure 4.** RoxS uses C-rich region 3 (CRR3) to interact with the extreme 5'-end of *yflS* mRNA. (**A**) Base-pairing between *yflS* and RoxS, or *yflS^mut5'* and RoxS^CRR3-2 compensatory mutants (**B**) In vitro degradation of the 5' end of *yflS* and *yflS^mut5'* transcripts by RNase J1 ± RoxS or the RoxS^CRR3-2 mutant. *yflS* and *yflS^mut5'* mRNAs were in vitro transcribed, dephosphorylated with CIP and radiolabeled at their 5'-end to obtain a 5' monophosphate extremities. Times after RNase J1 addition are indicated at the top of the gel. Reactions were performed at 15°C to facilitate interaction between *yflS^mut5'* and RoxS^CRR3-2 because the ΔG of the compensatory interaction is predicted to be weaker than the wt (−8.0 vs −13.7 kcal/mol). The migration positions of the *yflS* in vitro transcript (94 nt) and AMP are indicated. (**C**) Percent release of AMP from *yflS* and *yflS^mut5'* substrate RNAs by RNase J1 ± RoxS or the RoxS^CRR3-2 mutant as a function of time (minutes). The data shown correspond to the quantification of three independent experiments (technical replicates). Structure probing of RoxS ± *yflS* is shown in *Figure 4—figure supplement 1*. See also *Figure 4—source data 1*.

The following source data and figure supplement are available for figure 4:

**Source data 1.** Source data for the quantification of *yflS* and *yflS^mut* degradation by RNase J1 in vitro.

**Figure supplement 1.** RoxS uses C-rich region 3 (CRR3) to interact with the extreme 5'-end of *yflS* mRNA.

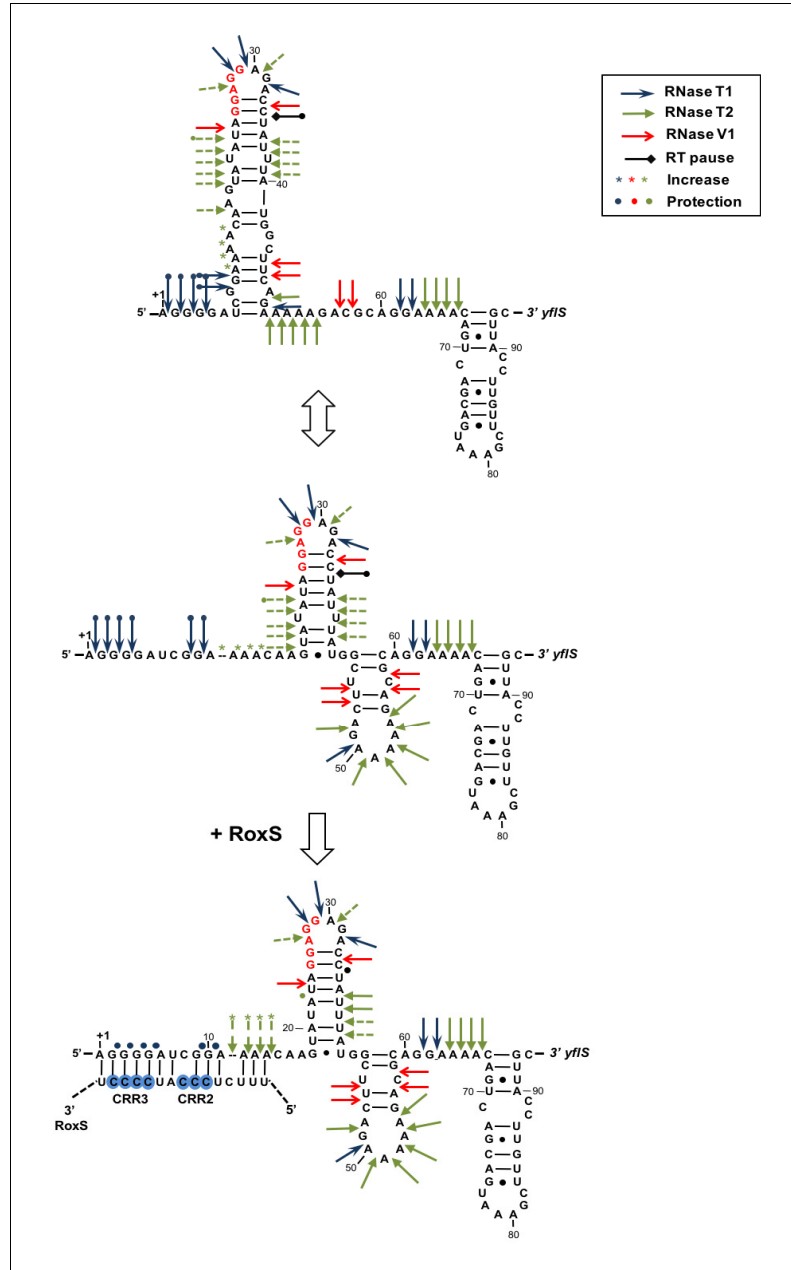

**Figure 5.** Modification of the *yflS* mRNA secondary structure upon RoxS binding. Summary of the cleavage patterns (*Figure 5—figure supplement 1*) of *yflS* alone (two potential equilibrium structures) or bound to RoxS generated by RNase V1 (red arrows), RNase T1 (blue arrows) and RNase T2 (green arrows). Weaker cleavages are represented by dashed arrows. The strong RT stop observed at position U35 (−5 relative to the AUG) is also indicated (black bar). Residues affected by the RoxS binding are indicated by colored asterisks (circles for inhibition; asteriks for stimulation of cleavage). The sequencing ladder was made by reverse transcribing RNA in the presence of dideoxynucleotides. The data suggests that a structural link exists between the 5' extremity and the downstream hairpin structure that contributes to the weak accessibility of the adenines 11 to 14 towards RNase T2 cleavages.

The following figure supplement is available for figure 5:

**Figure supplement 1.** Enzymatic probing of the *yflS* mRNA secondary structure in the presence and absence of RoxS.

the absence of RoxS, we also noticed a strong reverse transcriptase (RT) stop between C34 and U35 of the *yflS* leader. This RT stop is located between the SD and AUG start codon and was suppressed by the binding of the wild-type RoxS or the CRR1 mutant, but not the CRR3 variants. These results strongly support the prediction that RoxS base-pairs with the extreme 5′ end of the *yflS* mRNA through CRR3, and suggest that RoxS additionally induces a long-range perturbation of the mRNA structure close to the SD sequence.

Since the SD and the AUG codon are predicted to be poorly accessible and some structural changes close to the SD sequence were detected, we asked whether, in addition to protecting the *yflS* mRNA from degradation by RNase J1, RoxS might also have an impact on the translation of this mRNA. We therefore performed toeprinting experiments on the same 94-nt *yflS* leader fragment used for the RNase J1 in vitro degradation and structure probing assays (*Figure 5*). The *yflS* transcript was incubated with the 30S ribosomal subunit and initiator tRNA either with or without preincubation with RoxS (*Figure 6*). In the absence of RoxS, a characteristic reverse transcriptase stop was observed at position +16 relative to the start codon, showing that the translation initiation complex can be formed without RoxS. Addition of wild-type RoxS or the CRR1 mutant increased the signal at +16 approximately 3-fold, suggesting that RoxS stimulates the formation of the translation initiation complex. Significantly less stimulation was observed with the two CRR3 variants; it is possible that the apparent residual effect of these mutants comes from the extended base-pairing around CRR2 in these in vitro experiments. We also detected the strong RT stop between the SD and AUG start codon and, as previously, this species diminished with increasing concentration of wild type RoxS or CRR1 mutant, but not with the CRR3 mutants. The decrease of this signal between C34 and U35 is thus a signature of RoxS binding.

To confirm the effect of RoxS binding on *yflS* mRNA expression in vivo, we made a *yflS-lacZ* translational fusion in which the promoter, regulatory sequences and the first 436 amino acids (consisting of 11 out of 12 trans-membrane domains) of YflS were fused in frame to LacZ and inserted at the *amyE* locus of the chromosome. We measured the ß-galactosidase activity of this fusion in the presence and absence of RoxS, at different times after induction with malate (*Figure 7*). In both strains, ß-galactosidase activity increased progressively after the addition of malate to the growth medium. After 60 min induction, the specific ß-galactosidase activity was 2-fold higher in a WT background compared to the RoxS mutant strain, showing that RoxS stimulates YflS synthesis in vivo.

## RoxS blocks *yflS* degradation independently of translation

Since RoxS had an impact on both translation and degradation of the *yflS* mRNA (see above), it was conceivable that the primary effect of RoxS is on *yflS* mRNA translation. An increase of ribosome trafficking on the *yflS* mRNA could be sufficient to explain the increase in *yflS* stability as has been demonstrated for numerous *trans*-acting sRNAs in bacteria (*Lalaouna et al., 2013*). To uncouple the effect of degradation and translation, we mutated both the SD sequence (AGGAGA replaced by CAAAAA) and the fifth codon (AAA to TAA) to introduce a premature stop codon in the chromosomal copy of the *yflS* gene. This non-translated variant is called *yflS^mutTR^*. The half-life of *yflS^mutTR^* RNA was slightly decreased compared to the wild-type strain (1.6 min vs 2.5 min respectively; *Figure 8*, *Figure 8—figure supplement 1*), likely due to the decreased ribosome occupancy of the coding sequence and increased access to RNase Y. However, the *yflS^mutTR^* RNA was still destabilized about 2.3-fold in the absence of RoxS compared to the parental strain (1.6 min vs 0.7 min; *Figure 8*, *Figure 8—figure supplement 1*). These results suggest that the stimulation of translation by RoxS is not a prerequisite for the protection of *yflS* mRNA from RNase J1. In other words, RoxS-mediated stimulation of *yflS* mRNA translation and its protection from RNase J1 are independent phenomena.

## RoxS is transcriptionally repressed by Rex

As mentioned previously, the *yflS* gene encodes a malate transporter whose expression is induced by the two-component system MalKR when malate is present in the environment (*Tanaka et al., 2003*). Since RoxS post-transcriptionally regulates *yflS* gene expression, we asked whether carbon sources like glucose and malate had an impact on RoxS expression. To test this, we added 0.2% glucose and/or malate to the medium at mid-exponential phase in 2xTY medium and followed RoxS expression for 1 hr by Northern blot (*Figure 9A*). RoxS levels increased for about 30 min after addition of glucose before falling again at the 60 min time point. A similar profile, but of much greater

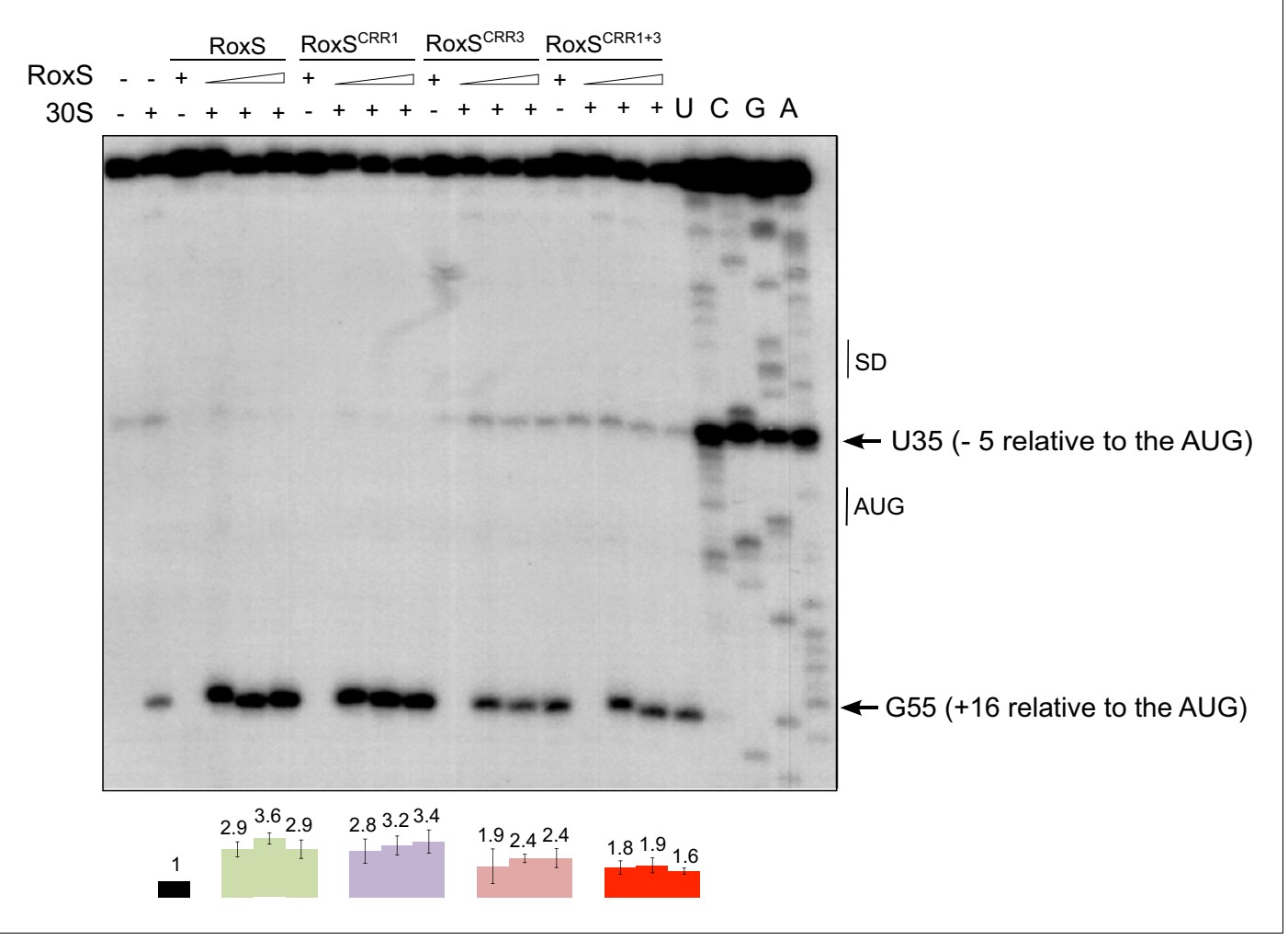

**Figure 6.** RoxS stimulates 30S binding to the *yflS* mRNA in vitro. Toeprint analysis of wild-type (WT) and various mutant forms of RoxS bound to 50 nM of the *yflS in vitro* transcript (94 nt). The toeprint formed by the 30S ribosomal subunit is indicated at +16 relative to the first nt of the AUG start codon. An additional RT stop at position U35 (−5 relative to the AUG), observed with RoxS variants containing a mutant CRR3 is also indicated. The Shine and Dalgarno sequence is indicated by SD. Three different quantities of RoxS (250, 500 and 1000 nM) were tested. The average stimulation of the toeprint at +16 by RoxS variants, normalized to the *yflS* mRNA alone, from three experiments (technical replicates) is presented under each lane of the autoradiogram, with standard errors as shown.

amplitude was seen upon addition of malate to the medium and addition of both carbon sources together (malate + glucose) had a cumulative effect on RoxS levels.

We performed a number of experiments to determine the regulator of RoxS expression in the presence of malate from among three potential candidates ResDE, MalKR and Rex. We had previously shown that RoxS is induced by nitric-oxide (NO) in the growth medium *via* the two-component system ResDE (*Durand et al., 2015a*). We therefore first tested the involvement of ResDE in the induction of RoxS expression by malate (*Figure 9B*). In the Δ*resDE* mutant strain, RoxS levels were lower than in the WT strain, as observed previously (*Durand et al., 2015a*). However, RoxS levels were still inducible by malate in the absence of ResDE, with the same kinetics as in the WT strain, showing that ResDE is not involved in this regulation.

The two-component system MalKR, known to respond to malate levels in *B. subtilis* and responsible for induction of *yflS* expression (see above), was also a good candidate for RoxS regulation by malate. In a Δ*malKR* strain mutant, RoxS levels no longer increased upon addition of malate (*Figure 9C*), suggesting a role for this two-component system in regulation. However, a comparison of the RoxS promoter sequence to those of the *maeN*, *maeA* and *yflS* genes, all known to bind

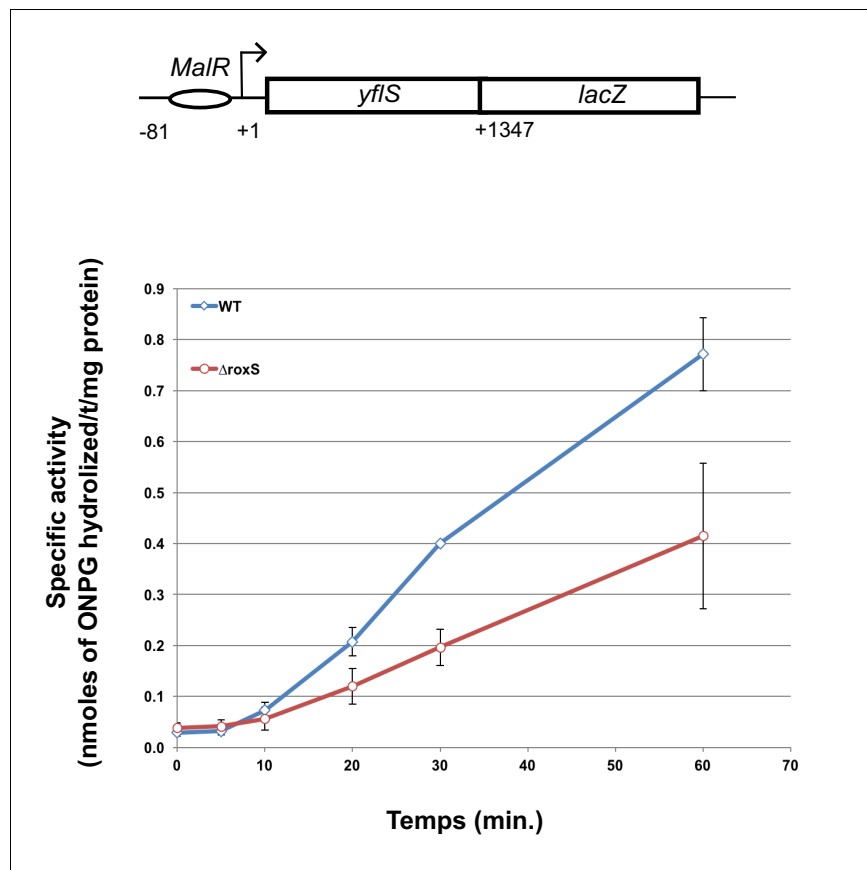

**Figure 7.** RoxS stimulates *yflS* translation in vivo. ß-Galactosidase assays of *yflS–lacZ* translational fusions in wild-type (WT) and Δ*roxS* strains. The reporter is under control of the *yflS* promoter and regulatory sequences, and consists of 436 amino acids of *yflS* coding sequence fused to *lacZ*. Malate (0.5%) was added at mid-exponential growth phase to induce the expression of the fusion. The levels of ß-galactosidase activity were assayed between 0 and 60 min after malate addition. The specific activity, in nmoles of ONPG hydrolyzed/mg protein/min, was determined in three independent experiments. Error bars indicate the standard deviation for each point. See also *Figure 7—source data 1*.

The following source data is available for figure 7:

**Source data 1.** Source data for the quantification of ß-galactosidase activity of the *yflS-lacZ* fusion.

MalR, did not reveal any obvious similarities. We purified the MalR protein and performed an electrophoretic mobility shift assay (EMSA) with a DNA fragment corresponding to the RoxS promoter, with the *maeA* promoter as a positive control and a portion of the *yflS* coding region as a negative control (*Figure 9—figure supplement 1*). We confirmed binding of MalR to the *maeA* promoter but found no specific interaction with the RoxS promoter region. These data suggest that the effect of the MalKR two-component system on RoxS levels is indirect.

Further examination of the sequence around the RoxS transcription start site revealed the presence of a potential binding site for the Rex transcriptional repressor (*Härtig and Jahn, 2012*) immediately downstream of the transcription start site (*Figure 9—figure supplement 2*). We therefore examined the induction of RoxS expression upon addition of malate to a Δ*rex* strain. In the absence of Rex, RoxS levels were already high before addition of malate induction and accumulated to even higher levels over time (*Figure 9D*). Furthermore, we no longer observed the reduction in RoxS levels one hour after malate addition. This result suggests that Rex represses RoxS transcription and that addition of malate to the growth medium causes a transient release from this repression for at least 30 mins (see discussion). To demonstrate that Rex directly binds the RoxS promoter region, we performed EMSA experiments with the *roxS* and the lactate dehydrogenase (*ldh*) gene promoter

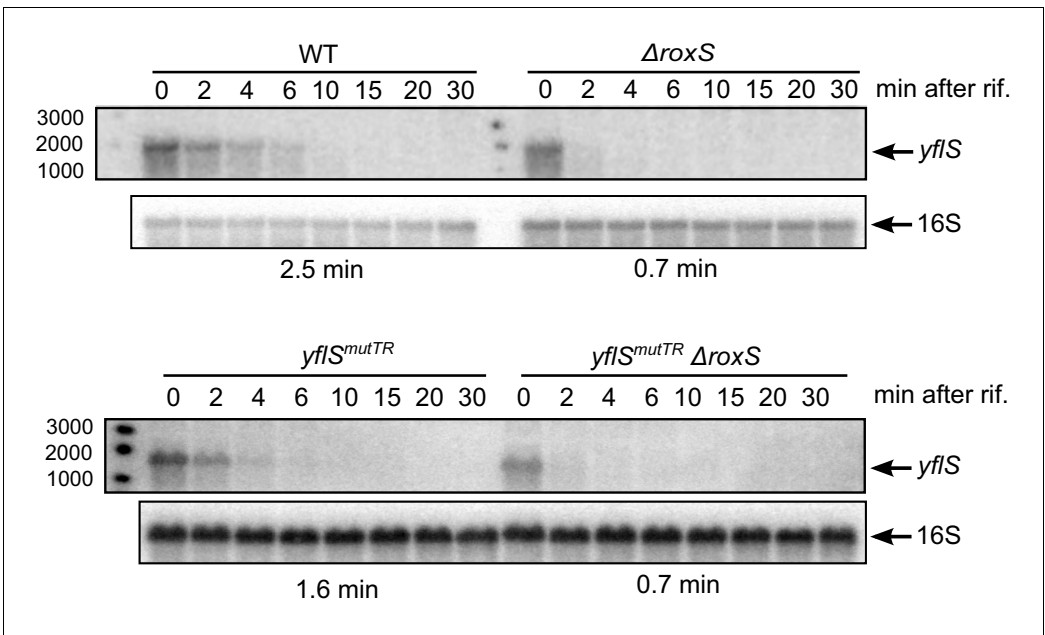

**Figure 8.** RoxS stabilizes a non-translated *yflS* mRNA variant in vivo. A representative Northern blot showing total RNA isolated at times after addition of rifampicin (Rif) to WT and Δ*roxS* strains, and *yflS^mutTR^* and *yflS^mutTR^* Δ*roxS* strains grown in the presence of 0.5% malate. Rifampicin was added at mid-exponential phase (OD$_{600}$ = 0.6). Half-lives calculated from two independent experiments (biological replicates) are given under each autoradiogram. The quantification of the Northern blots is presented in *Figure 8—figure supplement 1*.

The following source data and figure supplements are available for figure 8:

**Figure supplement 1.** RoxS stabilizes a non-translated *yflS* mRNA variant in vivo.

**Figure supplement 1—source data 1.** Source data for the quantification of *yflS^mutTR^* mRNA half-life.

sequences, and with a portion of the *yflS* coding region as a negative control (*Figure 10*). As expected, we did not observe complexes with the *yflS* coding region but we detected gel-shifted species with the *ldh* promoter, known to bind Rex (*Wang et al., 2008*). We also obtained a gel-shifted species for the *roxS* promoter fragment, confirming that Rex can directly bind this promoter region and explaining how it can inhibit transcription. Interestingly, the Rex protein and its predicted binding site just downstream of the *roxS* promoter are conserved in *S. aureus* and in Staphylococcaceae in general (*Pagels et al., 2010*), suggesting similar regulation may occur in these pathogenic bacteria (*Figure 9—figure supplement 2*).

## RoxS regulates *ppnkB* and *sucC* mRNA stability far more efficiently in the presence of malate

In our previous studies, we showed that RoxS blocks translation of the *ppnkB* and *sucC* mRNAs and indirectly destabilizes these transcripts (*Durand et al., 2015a*). Since RoxS expression is derepressed in the presence of malate, we retested the impact of RoxS on these two mRNAs in these growth conditions. We measured the half-lives of the *ppnKB* and *sucCD* mRNAs in the presence or in absence of malate by Northern blotting of RNAs isolated at different times after rifampicin addition. We previously showed that RoxS destabilized the *ppnkB* about 2-fold and had little impact on *sucCD* mRNA half-life in 2xTY with no added carbon source (*Figure 11A* and [*Durand et al., 2015a*]). In 2xTY plus malate, however, RoxS destabilized the *ppnkB* mRNA about 20-fold and the *sucD* mRNA about 8-fold (*Figure 11B*). This result confirms our previous data and shows that RoxS has a far greater impact on gene expression in the presence of malate.

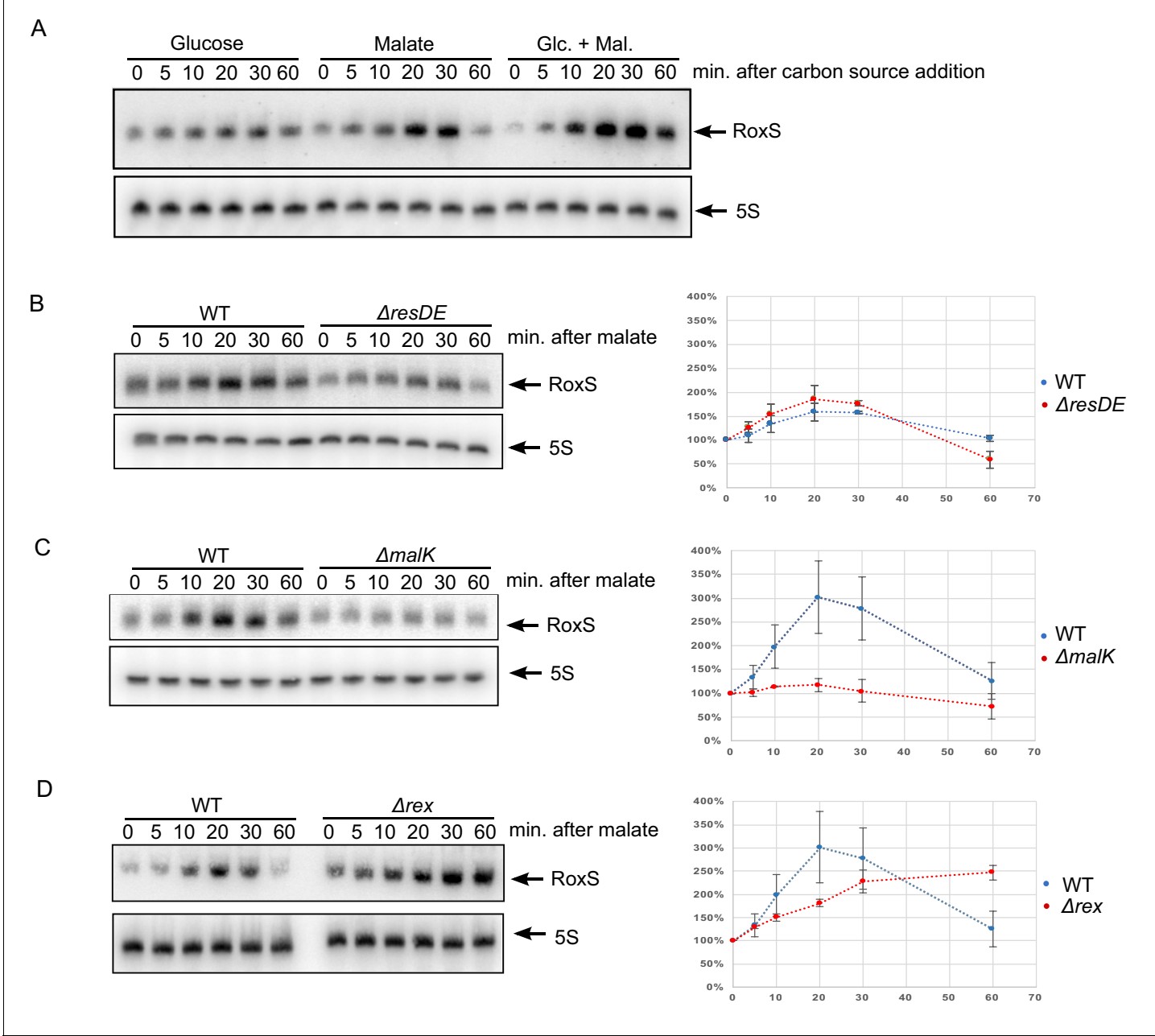

**Figure 9.** Derepression of RoxS expression in strains lacking Rex. (**A**) Northern blot of total RNA extracted from *B. subtilis* W168 (wild-type) before (time 0) and 5, 10, 20, 30 and 60 min after addition of 0.2% malate, 0.2% glucose, or both. (**B**) Northern blot of total RNA extracted from wild-type and Δ*resDE* strains before (time 0) and 5, 10, 20, 30 and 60 min after addition of 0.5% malate. (**C**) Same legend as panel B using wild-type and Δ*malK* strains. (**D**) Same legend as panel B using wild-type and Δ*rex* strains. Quantifications of the Northern blots are from two independent experiments (biological replicates). MalR binding assays are shown in *Figure 9—figure supplement 1*. Conservation of the Rex binding site in different *Staphylococci* and *Bacilli* RoxS promoters is presented *Figure 9—figure supplement 2*. See also *Figure 9—source data 1*.

The following source data and figure supplements are available for figure 9:

**Source data 1.** Source data for the quantification of RoxS derepression by malate in *Bacillus subtilis* mutant strains (Δ*resDE*, Δ*malK*, Δ*rex*).
**Figure supplement 1.** MalR does not bind the RoxS promoter.
**Figure supplement 2.** Alignment of RsaE/RoxS sequences from different *Staphylococci* and *Bacilli* adapted from *Durand et al. (2015a)*.

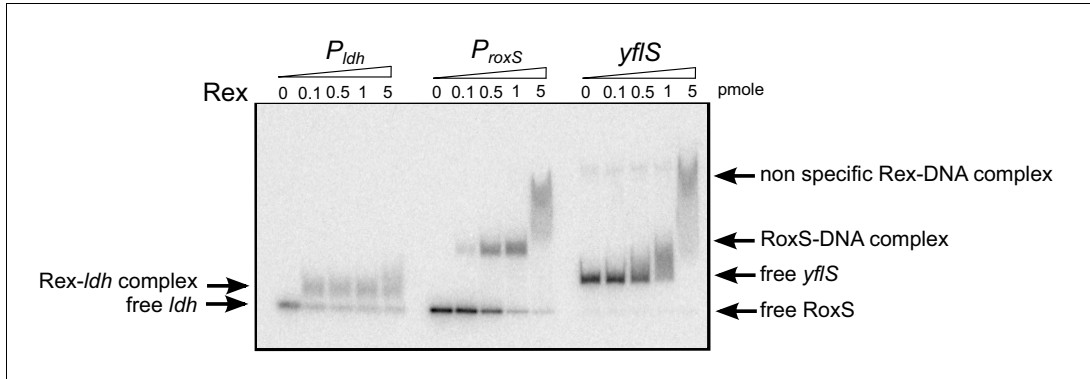

**Figure 10.** Rex binds the RoxS promoter region in vitro. Electrophoretic mobility shift assays (EMSA) of $^{32}$P-labelled double-stranded DNA fragments (0.1 pmol) containing the *ldh* Rex operator, the RoxS promoter or an internal sequence of *yflS* coding sequence. The quantities of Rex in the binding reaction are indicated (in pmoles) above each lane. EMSAs were performed three times (technical replicates).

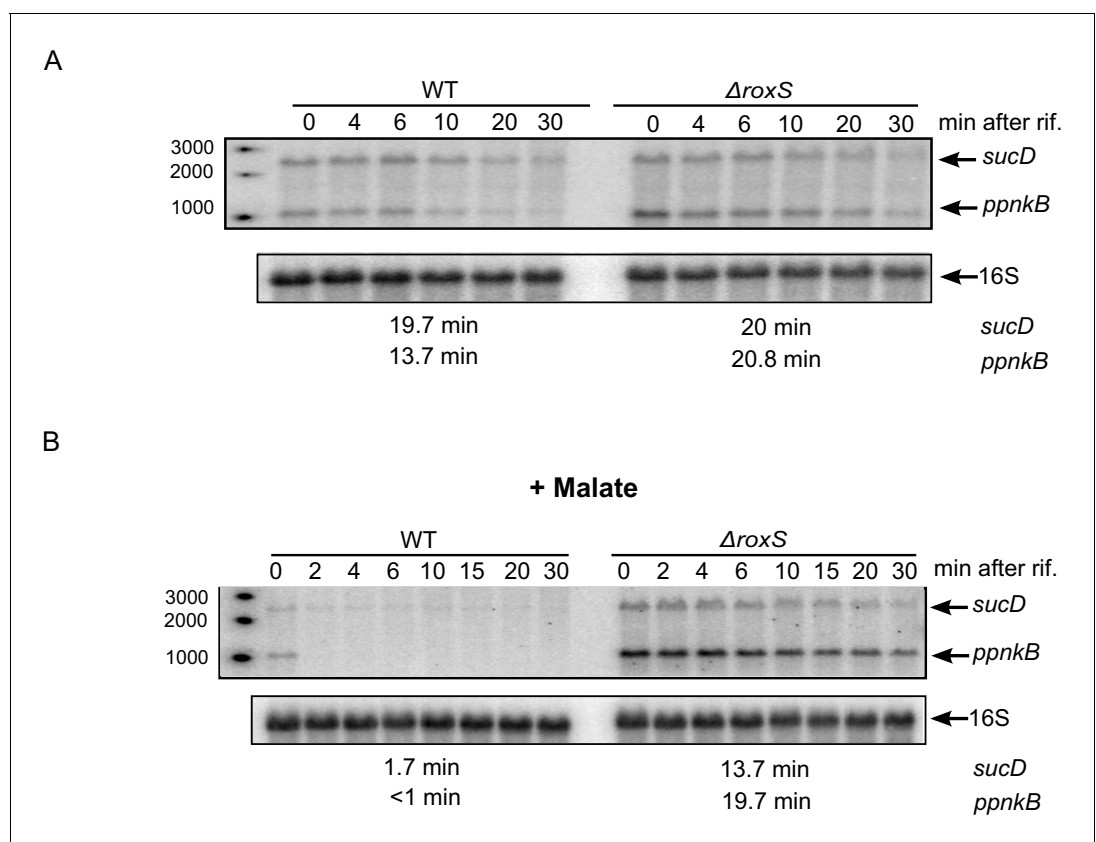

**Figure 11.** The effect of RoxS on *ppnKB* and *sucCD* mRNA stability is amplified in the presence of malate. (**A**) Northern blot of total RNA isolated at times after addition of rifampicin (rif) to WT and Δ*roxS* strains. Rifampicin was added at mid-exponential phase (OD$_{600}$ = 0.6). The half-lives of the *ppnkB* and *sucCD* mRNAs are indicated below the blot. (**B**) Same legend as panel A but with 0.5% malate in the medium. Northerns were performed twice (biological replicates).

## Discussion

In this study, we report for the first time that a small regulatory RNA (RoxS) can directly prevent 5'-to-3' mRNA degradation of its mRNA target. RoxS binds to the 5' end of the *yflS* mRNA through base-pairing interactions primarily with the C-rich region CRR3 (*Figure 1*). The inhibitory effect on RNase J1 5'−3' exoribonuclease activity can easily be understood in structural terms; the RNA binding channel that leads to the RNase J1 catalytic site is only long enough to accommodate 4–5 nts of single stranded RNA. The unbound *yflS* mRNA is predicted to have at least six unpaired nts before the first hairpin structure (*Figure 5*) and would thus be expected to be a good substrate for RNase J1 once the 5'-triphosphate group has been removed. When bound to RoxS, the duplexed 5' end of the *yflS* RNA can no longer reach the catalytic site, explaining how it is protected from degradation.

To our knowledge, there are only three other examples of direct mRNA stabilization by a *trans*-acting regulatory RNA in Gram-positive bacteria. In *Streptococcus mutans*, the 5'-UTR of the intact *irvA* mRNA interacts with the coding sequence of the *gbpC* mRNA, encoding a surface lectin implicated in dextran dependent aggregation stress (DDAG). This interaction blocks endoribonucleolyic cleavage of the *gbpC* mRNA by RNase J2 (*Liu et al., 2015*). *Obana et al. (2010, 2013)* showed that the *Clostridium perfringens colA* mRNA (encoding collagenase) is stabilized by the VR-RNA. Binding of the sRNA to the *colA* mRNA leads to a cleavage its 5'-UTR, which releases the *colA* SD sequence and stimulates translation initiation. The stabilization of *colA* depends on the binding of a ribosome to the SD sequence, but translation has no impact on mRNA stability. In the pathogen *Streptococcus pyogenes*, the FasX sRNA stabilizes the streptokinase encoding mRNA (*ska*) (*Ramirez-Peña et al., 2010*) by binding to its extreme 5' end *via* a sequence resembling that of the RoxS CRR3 (four consecutive C's matched with four consecutive G's). Our data would suggest that RNase J1 is a prime candidate for FasX regulation of the *ska* mRNA. In fact, the sRNA-dependent 5' end protection mechanism discovered here could in theory be used to activate expression of any mRNA in any microorganism by shielding it from enzymes (e.g. RppH, RNase J1, RNase E) requiring a single stranded 5' extremity to initiate degradation.

Our data also indicate that binding of RoxS to the 5' end of the *yflS* mRNA both protects it from degradation by RNase J1 (*Figures 2, 3* and *4*) and stimulates 30S binding to the SD sequence in vitro (*Figure 6*), suggesting an activation of translation. Moreover, the increased expression of the *yflS(436)-lacZ* fusion in the presence of RoxS is consistent with an increase in translation efficiency (*Figure 7*). We were able to uncouple the effect of RoxS on *yflS* mRNA stability and translation by making a non-translated variant (*yflS^{mutTR}*) and showed that RoxS is still able to stabilize this mRNA independently of translation (*Figure 8*). However, in all likelihood, the two events are indissociable in vivo with the wild-type *yflS* mRNA, that is, stabilization of the mRNA and increased translation go hand in hand to produce more of the YflS transporter in the presence of malate.

The enzymatic probing of *yflS* mRNA indicated that the binding of RoxS, in addition to protecting the 5' end from RNase T1 cleavage, causes subtle modifications of the secondary structure of the *yflS* leader, that could account for the stimulation of initiation complex formation. Most noticeably, a strong RT stop between C34 and U35 (five nucleotides upstream of the AUG) is lost upon binding of WT RoxS or the CRR1 mutant, but is retained when *yflS* mRNA is incubated with the inactive CRR3 variant. This suggests a structural modification to the apex of the stem-loop in the vicinity of the SD sequence upon binding to RoxS. Surprisingly, however, no other nucleotides in this neighborhood became more reactive to single stranded probes (e.g. RNase T1) in the presence of RoxS, in particular the two G-C pairs (25G/34C and 26G/33C) that are predicted to close the apical loop (*Figure 5*). This suggests that the 3D structure of *yflS* leader and the impact of RoxS may be more complex than the model presented in *Figure 5*. The *yflS* mRNA sequence contains several stretches of consecutive G-residues in the first 50 nts that could potentially form G-quadruplex RNA structures (*Kikin et al., 2006*). Alternatively, part of the 5' leader including the AAACAA sequence (nts 12–17) of *yflS* mRNA could potentially form a triple helix with the stem of the adjacent hairpin rich in AU pairings. In both cases, RoxS binding would disrupt a tertiary fold favoring the recognition of the ribosome. Our structure probing data are consistent with a model in which the duplex formed by binding of RoxS to the 5' end of the *yflS* mRNA not only blocks RNase J1 access but also disrupts a particular fold of the 5'UTR that is unfavorable to ribosome recruitment.

Our study showed that the expression of the RoxS sRNA is not only induced by nitric oxide (NO) as we described previously (*Durand et al., 2015a*), but also by the presence of malate in the growth

medium (*Figure 9*). The malate-induced increase in RoxS expression is indirectly dependent on the two-component system MalKR and directly dependent on release from Rex repression (*Figures 9* and *10*). The Rex repressor is a sensor of the $NAD^+$/NADH ratio in the cell. It has a 20,000-fold higher affinity for NADH than for $NAD^+$ and the binding of NADH to Rex provokes a structural rearrangement which prevents its binding to target promoters and allows the expression of genes involved in fermentation (*Larsson et al., 2005*; *Wang et al., 2008*; *McLaughlin et al., 2010*). The temporary release from Rex-mediated repression of RoxS expression may be linked to the generation of NADH in the early steps of malate metabolism, either in the conversion of malate to oxaloacetate by malate dehydrogenase (Mdh) in the TCA cycle, or its conversion to pyruvate by the malic enzymes MaeA, MalS and MleA (*Doan et al., 2003*; *Lerondel et al., 2006*; *Meyer and Stülke, 2013*) (*Figure 12*). The effect of the MalKR mutation on RoxS expression (*Figure 9C*) is most likely explained by the decrease in expression of the MalKR regulon, consisting of the malate transporter genes *yflS* and *maeN,* and that of the malic enzyme *maeA,* and the consequent impact on NADH levels and Rex activity.

*B. subtilis* has at least four malate transporters: MaeN (*Wei et al., 2000*) and YflS (*Tanaka et al., 2003*), both regulated by the two-component system MalKR (*Tanaka et al., 2003*), and MleN (*Wei et al., 2000*) and CimH, the latter of which is primarily a transporter of citrate (*Sobczak and Lolkema, 2005*). Malate is a common source of carbon for *B. subtilis* in its natural habitat since it is released by the roots of many plants (*Rudrappa et al., 2008*). Most of the malate imported by *B. subtilis* cells serves in gluconeogenesis and the excess is sent towards overflow metabolism/fermentation and the production of lactate, acetate, acetoin and 2,3-butanediol (*Kleijn et al., 2010*) (*Figure 12*). This overflow metabolism/fermentation allows $NAD^+$ regeneration, primarily catalyzed by lactate dehydrogenase (Ldh/LctE) that converts pyruvate to lactate (*Cruz Ramos et al., 2000*) (*Figure 12*). Transcription of the *maeN* mRNA, encoding the main malate transporter, is essentially turned off during fermentation and ethanol stress (*Nicolas et al., 2012*). It is possible that RoxS specifically stabilizes the *yflS* mRNA to compensate for this decrease in *maeN* expression, thus facilitating the continued uptake of malate during conditions where the fermentation/overflow pathway is turned on.

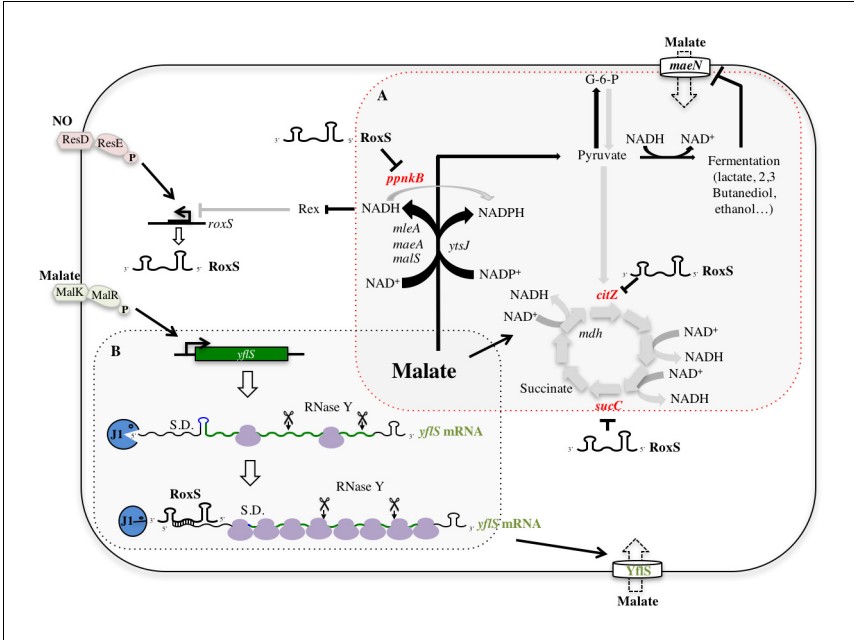

**Figure 12.** Model of various roles of RoxS upon addition of malate to culture medium. For details see text. Panel A shows reactions affected by RoxS that impact $NAD(P)^+$/NAD(P)H balance. Panel B shows the role of RoxS in *yflS* mRNA degradation. Targets negatively regulated by RoxS are in red; targets positively regulated by RoxS are in green.

The identification of Rex as a regulator of RoxS expression significantly expands the Rex regulon and contributes to our understanding of the pivotal role of Rex in Bacilli (*Laouami et al., 2014*); up to now only a handful of fermentation pathway genes or operons were known to be directly repressed by this transcriptional regulator. Our previous study showed that 30% of genes with increased expression in a $\Delta roxS$ strain are linked to oxido-reduction processes (*Durand et al., 2015a*). Its down-regulated targets include *ppnKB,* encoding NAD-kinase, and the TCA cycle components *sucCD* and *citZ*. PpnKB and the TCA cycle are significant contributors to the production of NADPH and NADH, respectively (*Figure 12*). These observations, together with the fact that RoxS is regulated by the NADH-sensitive transcriptional repressor Rex, suggest that a key role of RoxS is to temporarily turn down part of the TCA cycle to help restore the $NAD^+$/NADH balance. In the presence of malate, this re-equilibration is further aided by the increased activity of Ldh regenerating $NAD^+$ (see above). Once the $NAD^+$/NADH ratio is restored, Rex would be predicted to become active again as a repressor, explaining the decrease in RoxS levels seen in Northern blots about one hour after addition of malate and the loss of this down-regulation in the Rex mutant (*Figure 9D*). In further agreement with a role for RoxS in managing NADH levels, we showed in our previous report that expression of the lactate dehydrogenase gene (*ldh*) and the *cydA* gene, encoding a subunit of cytochrome bd, is up-regulated in $\Delta roxS$ cells (*Durand et al., 2015a*). Neither of these mRNAs is predicted to be a direct target of RoxS, but both are members of the Rex regulon. Their increased expression is thus most likely indicative of higher intracellular NADH levels and Rex derepression in the $\Delta roxS$ strain. Interestingly, a link between NO levels and regulation of virulence via NADH and Rex has been proposed in *S. aureus* (*Pagels et al., 2010*). The conservation of the Rex binding site downstream of the RsaE/RoxS promoter in *S. aureus* suggests that this pathway is evolutionarily conserved to best serve the adaptation of *B. subtilis* and *S. aureus* to their respective niches.

## Materials and methods

### Strains and constructs

Strains and oligonucleotides used are provided in *Supplementary files 1* and *2*, respectively.

Strain CCB803 (*malKR::spc*) was constructed by transforming *B. subtilis* W168 (SSB1002) with a PCR fragment generated by re-amplifying (oligo pair CC1722/CC1716) three overlapping PCR fragments corresponding to upstream (oligo pair CC1715/CC1717), downstream (oligo pair CC1719/CC1721) regions of the *malKR* operon and the spectinomycin resistance cassette (oligo pair CC1718/CC1720) from plasmid pDG1727 (*Guérout-Fleury et al., 1995*).

Strain CCB952 (*rex:pMUTIN ery*) was constructed by transforming the *B. subtilis* W168 strain with chromosomal DNA from strain MGNA-C201 from the Japanese National Institute of Genetics (NIG) collection.

The *yflS*(436)-*lacZ* fusion was constructed using a PCR fragment generated by amplifying the *yflS* gene (oligo pair CC1734/CC1789) including 436 amino acids of the coding sequence (−81 to +1347 relative to transcription start site) and cloned between the *Eco*RI and *Sal*I restriction sites of the integrative plasmid pHM2 (*Gendron et al., 1994*). *B. subtilis* W168 and CCB485 (*roxS::kan*) strains (*Durand et al., 2015a*) were transformed with pHM2-*yflS*(436)-*lacZ* yielding strains CCB831 and CCB832, respectively.

Strain CCB1022 (*yflS^mutTR*) was made by markerless mutation of the Shine-Dalgarno sequence and fifth codon of *yflS* using plasmid pMAD-yflS^mutTR according to *Arnaud et al., 2004*. Plasmid pMAD-yflS^mutTR was constructed as follows. Overlapping upstream and downstream fragments containing the mutation in *yflS* SD region (AGGAGA to CAAAAA) in the upstream fragment and the stop codon mutation in the downstream fragment were amplified using oligo pairs CC1656/2050 and CC2047/1657, respectively. The overlapping fragments were then assembled in a new PCR reaction with CC1656/1657, digested with *Sal*I and cloned in pMAD. Strain CCB1027 (*yflS^mutTR $\Delta roxS$*) was made by transformation of strain CCB1022 with chromosomal DNA of CCB485 (*roxS::kan*).

### RNA isolation and northern blots

RNA was isolated from mid-log phase *B. subtilis* cells growing in 2×YT medium either by the glass beads/phenol method described in *Bechhofer et al., 2008* or by the RNAsnap method described in *Stead et al., 2012*. Northern blots were performed as described previously (*Durand et al., 2012*).

The *yflS* riboprobe was transcribed in vitro using T7 RNA polymerase (Promega) and labeled with [α-$^{32}$P]-UTP using a PCR fragment amplified with oligo pair CC1566/CC1567 as template.

## Mapping of 5' ends by primer extension assays following Xrn1 digestion

Primer extension assays were performed as described previously on total RNA (*Britton et al., 2007*) with the *yflS* oligo (CC1537) and *hbs* oligo (CC463) 5' end-labeled with T4 polynucleotide kinase (PNK) and [γ-$^{32}$P]-ATP. Xrn1 depletion of 5' monophosphorylated RNA was performed as described in *Sinturel et al. (2015)*. Briefly, 15 µg of total RNA was digested for 1 hr at 37°C with 2 units of XrnI in NEB buffer 3. Total treated and non-treated RNAs were phenol extracted and ethanol precipitated. The efficiency of digestion was tested by running total RNA on an agarose gel to visualize digestion of ribosomal RNA.

## RNase J1 assays

The 94-nt *yflS* and *yflS$^{mut5'}$* RNA fragments were transcribed in vitro from PCR fragments containing the leader region and the first 55 nts of the *yflS* coding sequence. The *yflS* and *yflS$^{mut5'}$* templates were amplified with the oligo pairs CC1658/CC1537 and CC1537/CC2040 respectively. RoxS and its mutant derivatives (CRR1, CRR3 and CRR1+3) were made similarly using PCR fragments amplified from their respective plasmids 639, 640 and 641, using oligos CC1832/CC1833 (*Durand et al., 2015a*). The CRR3-2 mutant was made by overlapping PCR. The CCCCU sequence of CRR3 was mutated to UUCUC using oligo pairs CC1832/CC2042 and CC1833/CC2041 and reamplified with CC1832 and CC1833. The *yflS* RNA was dephosphorylated with Calf Intestinal Phosphatase (CIP) (10 U/µl; Biolabs) and 5' end-labeled with T4 polynucleotide kinase (PNK) and [γ-$^{32}$P]-ATP. The *yflS* RNA (1.2 pmol), RoxS and its mutant derivatives (3 pmol) were pre-heated separately 5 min at 95°C in water or transcription buffer, respectively, and slowly cooled to 20°C. The *yflS* RNA was mixed with RoxS or its mutant derivatives in 1X J1 buffer (20 mM Tris pH 6.8, 8 mM MgCl2, 100 mM NH4Cl, 0.1 mM DTT). The reaction was incubated at 37°C or at 15°C (as indicated) and stopped with RNA loading dye (Ambion) before (time 0) and 2, 4, 6 and 10 min after addition of RNase J1 (0.5 µg per reaction) at 37°C, and 30 and 60 min after addition of RNase J1 at 15°C. RNA was also incubated without RNase J1 for 10 min at 37°C or 60 min at 15°C as a control.

## Toeprinting assays

The *yflS* RNA fragments were transcribed in vitro as described for RNase J1 assays. The transcribed RNAs were purified from 8% polyacrylamide-8M urea gels. After elution in 0.5 M ammonium acetate/1 mM EDTA buffer, the RNAs were precipitated twice with ethanol. The 5' end-labeling of dephosphorylated DNA oligonucleotides was performed with T4 PNK and [γ-$^{32}$P]-ATP. Before use, RNAs were denatured/renatured by incubation at 90°C for 2 min in the absence of magnesium and salt, then 1 min on ice, followed by an incubation step at 37°C for 15 min in TMN buffer (20 mM Tris-acetate pH 7.5, 10 mM magnesium acetate, 150 mM sodium acetate). Toeprints were performed with *E. coli* 30S subunits. The preparation of ribosomal subunits, the formation of a simplified translational initiation complex with mRNA and the primer extension conditions were performed according to *Fechter et al. (2009)*. Standard conditions contained 15–50 nM RNA transcript annealed to a 5' end-labeled oligonucleotide (CC1537: complementary to codons 11 to 18), 250 nM *E. coli* 30S ribosomal subunits, and 40 to 80 nM of RoxS in 10 µl final volume of buffer containing 20 mM Tris-acetate, pH 7.5, 60 mM NH$_4$Cl, 10 mM magnesium acetate, and 3 mM ß-mercaptoethanol. After 10 min at 37°C, the initiator tRNA (1 µM) was added and the reaction was incubated for a further 5 min at 37°C. Reverse transcription was performed with one unit of AMV reverse transcriptase for 15 min at 37°C.

## Probing of RNA secondary structure

Structure probing of RoxS was performed with wild-type RoxS labeled at its 5' end using [γ-$^{32}$P]-ATP and T4 polynucleotide kinase (Ambion). Complex formation was performed with 5' end-labeled RoxS and increasing concentrations of cold *yfl*S mRNA (0.5, 1 and 4 pmoles). The two RNAs were denatured at 90°C for 1 min in sterile H$_2$O and cooled on ice for 1 min; hybridization was at 20°C for 15 min in 1xTMN buffer (20 mM Tris-HCl pH 7.5, 10 mM MgCl$_2$, 150 mM KCl). Enzymatic hydrolysis

was performed on 5' end-labeled RoxS (50,000 cps) either alone or bound to *yfl*S mRNA (final concentrations of 50, 100, 400 nM) in the presence of 1 µg carrier tRNA in a total volume of 10 µl containing TMN buffer. Reactions were done at 20°C for 5 min with RNase V1 (0.008 U, Ambion), or for 10 min with RNase T2 (0.025 U, MoBiTech). Hydrolysis was stopped by phenol extraction followed by RNA precipitation. The 5' end-labeled RoxS fragments were migrated on 8% polyacrylamide/8 M urea gels.

Structure probing of *yfl*S mRNA was performed with the 94 nts *yfl*S transcript (see RNase J1 assays). Complex formation between *yfl*S mRNA and increasing concentrations of RoxS sRNA was carried out as follows. Both RNAs were annealed under the same conditions as described just above. The reactions were performed on *yfl*S (0.5 pmole) either alone or bound to RoxS in a total reactional volume of 10 µL in TMN Buffer 1X, in the presence of 1 µg carrier tRNA. Enzymatic hydrolysis was performed at 20°C for 5 min with 0.004 U of RNase V1 (Ambion) or 0.5 U of RNase T1 (Thermo Scientific), and at 20°C for 10 min with 0.025 U of RNase T2 (MoBiTech). The reactions were stopped by precipitation of the RNA by adding 20 µL of a solution containing 1M guanidine thiocyanate, 0.167% N-lauryl sarcosine, 10 mM DTT and 83% isopropanol. The enzymatic cleavages were detected by primer extension using an end-labeled oligonucleotide (CC1537) and reverse transcriptase as previously described (*Helfer et al., 2014*). The cleavage sites were analyzed after fractionation of the DNA fragments on a 10% polyacrylamide-8 M urea gel-TBE 1X followed by autoradiography.

## Electrophoretic mobility shift assays (EMSA)

For EMSA assays with Rex, a DNA fragment containing the *ldh* gene Rex operator was prepared by annealing two oligonucleotides LDH2F/LDH2R (a kind gift from C. von Wachenfeldt) (*Supplementary file 1*) in binding buffer (100 mM NaCl and 10 mM Tris, pH 8). DNA fragments containing the RoxS promoter and an internal sequence of the *yfl*S ORF were synthetized by PCR using the oligo pairs CC1859/CC1860 and CC1578/CC1768, respectively. Oligos were radiolabelled using PNK and [γ-$^{32}$P]-ATP. Binding reactions (20 µl) were performed in 10 mM potassium phosphate buffer (pH 7.5), 10% glycerol, 1 µg herring sperm DNA as non-specific competitor, and contained approximately 0.1 pmol of labelled DNA.

For EMSA assays with MalR, a DNA fragment containing the *maeA* MalR operator was synthetized by PCR using the oligo pair CC1861/CC1862. DNA fragments containing the RoxS promoter and an internal sequence of *yfl*S ORF were prepared as described above for the EMSA experiment with Rex. Oligos were radiolabelled using PNK and [γ-$^{32}$P]-ATP. Binding reactions (20 µl) were performed in 10 mM Tris-HCl buffer (pH 8), 50 mM NaCl, 1 mM EDTA, 1 mM DTT, 5% glycerol, 1 µg herring sperm DNA as non-specific competitor, and contained approximately 0.5 pmol of labeled DNA.

All EMSA reactions were incubated for 15 min at 30°C then loaded onto a native 6% polyacrylamide gel in TBE. Gels were run at 4°C at 150 V, dried and analyzed by Phosphor Imager.

## Acknowledgements

We thank Claes von Wachenfeldt for the kind gift of the *B. subtilis* Rex protein and oligonucleotides corresponding to the *ldh* operator, and Matthieu Jules for the MalR (YufM) overexpressing strain. We thank M Springer, S Aymerich, M Jules, T Doan and E Westhof for helpful discussions.

## Additional information

### Funding

| Funder | Grant reference number | Author |
| --- | --- | --- |
| Agence Nationale de la Recherche | ANR-16-CE12-0002-01 BaRR | Sylvain Durand |
| Centre National de la Recherche Scientifique | UMR8261 | Sylvain Durand Frédérique Braun Anne-Catherine Helfer Pascale Romby |

| | | Ciarán Condon |
|---|---|---|
| Centre National de la Recherche Scientifique | UPR9002 | Sylvain Durand<br>Frédérique Braun<br>Anne-Catherine Helfer<br>Pascale Romby<br>Ciarán Condon |
| Agence Nationale de la Recherche | ANR-10-LABX-0036<br>NETRNA | Pascale Romby |
| Université de Strasbourg | UPR9002 | Pascale Romby |
| Université Paris Diderot | UMR8261 | Ciarán Condon |
| Agence Nationale de la Recherche | ANR-12-BSV6-0007<br>asSUPYCO | Ciarán Condon |

The funders had no role in study design, data collection and interpretation, or the decision to submit the work for publication.

### Author contributions

SD, Conceptualization, Formal analysis, Funding acquisition, Validation, Investigation, Methodology, Writing—original draft, Project administration, Writing—review and editing; FB, Conceptualization, Formal analysis, Investigation, Writing—review and editing; A-CH, Formal analysis, Investigation, Methodology; PR, CC, Conceptualization, Formal analysis, Supervision, Funding acquisition, Validation, Visualization, Project administration, Writing—review and editing

### Author ORCIDs

Sylvain Durand, http://orcid.org/0000-0003-3509-6463

# Additional files

### Supplementary files

• Supplementary file 1. Strains used in this study.

• Supplementary file 2. Oligos used in this study.

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
