## [Decision Letter]

Thank you for submitting your article "sRNA-mediated activation of gene expression by inhibition of 5'-3' exonucleolytic mRNA degradation" for consideration by *eLife*. Your article has been favorably evaluated by Richard Losick (Senior Editor) and three reviewers, one of whom is a member of our Board of Reviewing Editors. The following individual involved in review of your submission has agreed to reveal his identity: David Bechhofer (Reviewer #3).

The reviewers have discussed the reviews with one another and the Reviewing Editor has drafted this decision to help you prepare a revised submission.

Summary:

This manuscript by Sylvain Duran et al. reports the discovery of 5'end-dependent post-transcriptional activation of a *Bacillus subtilis* mRNA by the conserved regulatory sRNA RoxS. The authors propose that RoxS recognizes the *yflS* mRNA at the very 5' end and thereby blocks 5' exonucleolytic degradation of this target by RNase J1. This would be the first time a bacterial sRNA has been shown to regulate gene expression in trans by inhibiting 5'-3' exonucleolytic degradation, the latter of which being similar to RNA degradation in eukaryotes. In the second part of this manuscript, the authors present experiments indicating that RoxS binding also remodels structurally the *yflS* mRNA to make it more competent for translation. In the third part, RoxS is shown to be transcriptionally controlled by the important DNA binding protein Rex. Additionally, experiments are presented to elucidate RoxS activity under different nutritional conditions. Specifically, the authors show that RoxS expression is negatively regulated by Rex, a transcriptional repressor that is itself indirectly regulated by the presence of malate. A regulatory scheme for malate metabolism involving RoxS-mediated positive and negative regulation could be developed from these results (Figure 11 is remarkable!).

Generally, the study is significant as much less is known about sRNA regulation in *B. subtilis* than in *E. coli*, and the *B. subtilis* suite of ribonucleases is considerably different from that of *E. coli* – most notably the presence of a 5' exoribonuclease (RNase J1), an activity that is not known to exist in *E. coli*. Furthermore, RoxS and regulatory sequences controlling its expression are conserved in other Gram-positives; thus, the current findings are likely broadly applicable.

The article is well written and the majority of the conclusions are well supported by the data. However, in its current form, the study lacks some key experiments and would benefit from a more thorough quantification of the data. For orientation, the first part of this manuscript, i.e. the discovery of sRNA-mediated mRNA activation by blocking 5' exo decay, clearly constitute sufficient novelty to make this a strong candidate for an *eLife* paper. However, it needs to be better developed. The second part – translational control – is presently not convincing. The third part constitutes an important part, too, since it puts the activity of RoxS in physiological context.

Essential revisions:

1) Subsection “Mapping of the 5’ extremity of the *yflS* mRNA” and Figure 2: While primer extension is an effective technology to map the 5' end of transcripts, it fails to determine the chemical nature of the 5' end. In other words, the determined 5' end might as well be that of a stable degradation intermediate. Differential 5' RACE experiments (see PMID: 8755555) would be one way to test this hypothesis. Alternatively, there seem to be ample published RNA-seq data for Bacillus for the authors; do they support this transcription start site? In addition, is the promoter conserved? In any case, this part on 5' end mapping seems rather confirmatory and doesn't deserve to be a main figure (move to the supplement).

2) Subsections “RoxS regulates *yflS* mRNA stability” and “RoxS blocks degradation of the *yflS* mRNA by RNase J1”: The control at the level of mRNA stability needs to be demonstrated more firmly. The authors should try to fully uncouple it from any translational control by using a non-translated 5' fragment of the *yflS* mRNA. This can be achieved by co-expression of RoxS with such a shorter RNA, perhaps as a fusion with an unrelated stable transcript. This has been done with other sRNAs that activate targets by increasing mRNA stability (SgrS, PMID:23582330; RydC, PMID:24141880). Such shorter fusion RNA target would also aid better mutational studies of the target.

3) Subsection “RoxS protects *yflS* mRNA from in vitro degradation by RNase J1”: The prediction that C-rich regions of RoxS are involved in base pairing with the *yflS* mRNA comes out of the blue. Explain what's known about other RoxS-target interactions. Is the predicted target site in *yflS* mRNA supported by conservation at the nucleotide level?

4) Subsection “RoxS protects yflS mRNA from in vitro degradation by RNase J1”: Here and elsewhere, the predicted base pairing interaction remains to be more firmly demonstrated. Use compensatory base pair changes in sRNA and target, or show RNA structure probing data of both native and mutant variants of the RNA partners.

5) Subsection “RoxS also activates translation of the *yflS* mRNA”, second paragraph and Figure 6: It is puzzling that the CCR1+3 variant of RoxS still affords an almost two-fold translational activation (30S binding) of the target, especially in light of the fact that i) wild-type RoxS manages a mere 3-fold activation in this assay, and that ii) the CCR3 mutation should nullify base pairing (according to Figure 1). This again demands a better definition of the RoxS-*yflS* RNA interaction in order to design more suitable mutants. Moreover, the 30S toeprint assay should include a second non-target mRNA to demonstrate that RoxS activity is specific.

6) Better quantification. Figure 4: It would be useful to quantify and plot the intensities of the bands corresponding to the 94 nt *yflS* transcript and the bands corresponding to AMP. Figure 7: the effect of RoxS on *yflS* in these experiments is approximately 2-fold, which probably presents the sum of transcript protection and translation activation. To discriminate the individual contributions of RoxS to this process, it would be useful to generate a *yflS* variant that is not translated (e.g. by mutating the start codon of the mRNA) and test the effect of RoxS on Northern Blots. Figure 8: Again, it would be useful to quantify the bands corresponding to RoxS in these experiments (e.g. by setting the 0 min time point to 100%). This would allow for a thorough comparison of RoxS expression across the presented experiments.

---

## [Author Response]

*Essential revisions:*

*1) Subsection “Mapping of the 5’ extremity of the yflS mRNA” and Figure 2: While primer extension is an effective technology to map the 5' end of transcripts, it fails to determine the chemical nature of the 5' end. In other words, the determined 5' end might as well be that of a stable degradation intermediate. Differential 5' RACE experiments (see PMID: 8755555) would be one way to test this hypothesis. Alternatively, there seem to be ample published RNA-seq data for Bacillus for the authors; do they support this transcription start site? In addition, is the promoter conserved? In any case, this part on 5' end mapping seems rather confirmatory and doesn't deserve to be a main figure (move to the supplement).*

To confirm that the mapped 5’end of *yflS* corresponds to a primary transcript, we performed a new reverse transcription reaction on total RNA treated (or not) with XRN1. This 5’-3’ exoribonuclease digests RNA with a 5’ monophosphate. We show that the mapped 5’end of *yflS* is resistant to digestion by XRN1, suggesting that this extremity is a 5’ triphosphate, whereas a known processed transcript of the *hbs* mRNA is susceptible to XRN1 treatment. As suggested, we moved this figure to the supplemental information section (Figure 1—figure supplement 1).

*2) Subsections “RoxS regulates yflS mRNA stability” and “RoxS blocks degradation of the yflS mRNA by RNase J1”: The control at the level of mRNA stability needs to be demonstrated more firmly. The authors should try to fully uncouple it from any translational control by using a non-translated 5' fragment of the yflS mRNA. This can be achieved by co-expression of RoxS with such a shorter RNA, perhaps as a fusion with an unrelated stable transcript. This has been done with other sRNAs that activate targets by increasing mRNA stability (SgrS, PMID:23582330; RydC, PMID:24141880). Such shorter fusion RNA target would also aid better mutational studies of the target.*

To uncouple mRNA stability from translational control of *yflS*, we constructed a strain in which the chromosomal copy of the yflS gene contained both a mutant SD sequence and a premature stop codon at the fifth codon (belt and braces approach!).The half-life of this *yflS^mutTR^*mRNA is a little shorter than the WT *yflS* mRNA (1.6 min vs. 2.5 min respectively; Figure 8 and Figure 8—figure supplement 1), as one might expect since translating ribosomes can no longer protect this mRNA from RNase Y. However, the *yflS^mutTR^* mRNA is still destabilized in absence of RoxS similar to the parental strains (Figure 8). This result nicely confirms that RoxS can stabilise the *yflS* mRNA independently of its effect on translation.

*3) Subsection “RoxS protects yflS mRNA from* in vitro *degradation by RNase J1”: The prediction that C-rich regions of RoxS are involved in base pairing with the yflS mRNA comes out of the blue. Explain what's known about other RoxS-target interactions.*

In a previous study, we showed that the C-rich region CRR3 of RoxS was involved in the interaction with the *ppnkB* mRNA target. We have better explained the rationale for focussing on the C-rich elements in the text (subsection “RoxS uses C-rich region 3 (CRR3) to interact with the extreme 5’-end of *yflS* mRNA”, first paragraph).

*Is the predicted target site in yflS mRNA supported by conservation at the nucleotide level?*

*Bacillus subtilis* and related bacteria have several di- and tri-carboxylate transporters homologous to YflS making it difficult to identify this particular transporter with certainty. For those examples where we can be relatively sure, because synteny is maintained, the 5'-UTR of *yflS* is highly conserved at the nucleotide level and, since the RoxS sRNA is identical to that of *B. subtilis* in these organisms, the potential base-pairing with CRR3 of RoxS is maintained. A figure showing an alignment of these 5'-UTRs and the complementary section of RoxS is shown in the supplementary section (Figure 1—figure supplement 2) and a statement describing this conservation is included in the manuscript (subsection “Mapping of the 5’ extremity of the *yflS* mRNA”)

*4) Subsection “RoxS protects yflS mRNA from in vitro degradation by RNase J1”: Here and elsewhere, the predicted base pairing interaction remains to be more firmly demonstrated. Use compensatory base pair changes in sRNA and target, or show RNA structure probing data of both native and mutant variants of the RNA partners.*

To confirm the base pairing prediction, we added 2 experiments to the paper.

In the first, we assayed the in vitro ability of RNase J1 to degrade mutant 5' fragments of the *yflS* mRNA (*yflS^mut5'^*), in which the predicted base-pairing with RoxS was disrupted and then restored with compensatory mutations in CRR3 (CRR3-2). In in vitro RNA degradation assays similar to those shown in Figure 3, we show that mutant RoxS (CRR3-2) shows weaker protection of wt *yflS* transcripts from RNase J1. Similarly, wt RoxS shows weaker protection of *yflS^mut5'^*, whereas when the RNAs bearing compensatory mutations are mixed (CRR3-2 and *yflS^mut5'^*), protection from RNase J1 degradation is recovered. This result confirms that the RoxS CRR3 element is involved in binding to the 5’-end of the *yflS* RNA. These data are included in a new figure (Figure 4).

We also probed the RoxS RNA secondary structure with RNase V1 and RNase T2 after addition of increasing concentrations of *yflS* RNA (Figure 3—figure supplement 1). This experiment showed increased cleavages by RNase V1 near CRR3 and increased protection from RNase T2 between CRR2 and CRR3 of RoxS upon addition of *yflS*, but no protection at the CRR1 sequence.

Together, these experiments demonstrate convincingly that RoxS uses the CRR2-3 region as predicted to interact with the *yflS* mRNA.

*5) Subsection “RoxS also activates translation of the yflS mRNA”, second paragraph and Figure 6: It is puzzling that the CCR1+3 variant of RoxS still affords an almost two-fold translational activation (30S binding) of the target, especially in light of the fact that i) wild-type RoxS manages a mere 3-fold activation in this assay, and that ii) the CCR3 mutation should nullify base pairing (according to Figure 1). This again demands a better definition of the RoxS-yflS RNA interaction in order to design more suitable mutants.*

We have done the toeprint experiment three times and the quantification represents the average and error bars for these three experiments. Visually the result is always the same: the wt and the CRR1 mutant give a much stronger toeprint than either the CRR3 or CRR1+3 mutant and the apparent residual effect of these mutants, particularly CRR1+3, compared to the (-)RoxS control appears stronger in the quantification than to the eye. Furthermore, the U35 RT stop which is lost upon efficient RoxS binding to *yflS* (even in the absence of ribosomes e.g. lane 3) behaves as expected for the CRR3 or CRR1+3 mutants. We are confident that we have identified the correct base-pairing, particularly with the additional experiments provided (compensatory mutations, structure probing of RoxS). It is possible that since the base pairing extends into the CRR2 region (validated by RNase T2 structure probing of RoxS described above) that this is sufficient to account for the residual toe-print stimulation in vitro (it is unlikely to be sufficient in vivo in a more competitive environment). We have included a sentence stating that "it is possible that the apparent residual effect of these mutants comes from the extended base-pairing around CRR2 in these in vitro experiments."

*Moreover, the 30S toeprint assay should include a second non-target mRNA to demonstrate that RoxS activity is specific.*

RoxS specificity was demonstrated in a previous paper. In Durand et al. *2015* (Figure 11),the toeprint experiment on the *sucC* mRNA shows that the full length of RoxS has strictly no impact on translation initiation of this mRNA.

*6) Better quantification. Figure 4: It would be useful to quantify and plot the intensities of the bands corresponding to the 94 nt yflS transcript and the bands corresponding to AMP.*

We have calculated the percent of AMP released compared to total signal (AMP + full-length transcript) and these graphs have been added to the Figure (now Figure 3).

*Figure 7: the effect of RoxS on yflS in these experiments is approximately 2-fold, which probably presents the sum of transcript protection and translation activation. To discriminate the individual contributions of RoxS to this process, it would be useful to generate a yflS variant that is not translated (e.g. by mutating the start codon of the mRNA) and test the effect of RoxS on Northern Blots.*

This has been done (also to address point no. 2) and has been included as a new Figure 8). It confirms that we can uncouple the two events.

*Figure 8: Again, it would be useful to quantify the bands corresponding to RoxS in these experiments (e.g. by setting the 0 min time point to 100%). This would allow for a thorough comparison of RoxS expression across the presented experiments.*

This has been done and added to the figure (now Figure 9).